# Coordinated hedgehog signaling induces new hair follicles in adult skin

Xiaoyan Sun[†], Alexandra Are[†], Karl Annusver[‡], Unnikrishnan Sivan[‡§], Tina Jacob, Tim Dalessandri, Simon Joost, Anja Füllgrabe[#], Marco Gerling, Maria Kasper*

Department of Biosciences and Nutrition, Karolinska Institutet, Huddinge, Sweden

**Abstract** Hair follicle (HF) development is orchestrated by coordinated signals from adjacent epithelial and mesenchymal cells. In humans this process only occurs during embryogenesis and viable strategies to induce new HFs in adult skin are lacking. Here, we reveal that activation of Hedgehog (Hh) signaling in adjacent epithelial and stromal cells induces new HFs in adult, unwounded dorsal mouse skin. Formation of de novo HFs recapitulated embryonic HF development, and mature follicles produced hair co-occurring with epithelial tumors. In contrast, Hh-pathway activation in epithelial or stromal cells alone resulted in tumor formation or stromal cell condensation respectively, without induction of new HFs. Provocatively, adjacent epithelial-stromal Hh-pathway activation induced de novo HFs also in hairless paw skin, divorced from confounding effects of pre-existing niche signals in haired skin. Altogether, cell-type-specific modulation of a single pathway is sufficient to reactivate embryonic programs in adult tissues, thereby inducing complex epithelial structures even without wounding.

*For correspondence:
maria.kasper@ki.se

[†]These authors contributed equally to this work
[‡]These authors also contributed equally to this work

Present address: [§]Kennedy Institute of Rheumatology, University of Oxford, Oxford, United Kingdom; [#]European Molecular Biology Laboratory, European Bioinformatics Institute, Hinxton, United Kingdom

Competing interests: The authors declare that no competing interests exist.

## Introduction

The number and pattern of hair follicles (HFs) are specified before birth in humans. In the mouse this is true for most body areas such as back skin (*Alonso and Rosenfield, 2003*; *Millar, 2002*; *Paus and Cotsarelis, 1999*). HF morphogenesis requires Hedgehog (Hh) and Wnt/β-catenin signaling and becomes first morphologically visible at embryonic day 14.5 (E14.5) in mice (*Chiang et al., 1999*; *Gat et al., 1998*; *Lo Celso et al., 2004*; *St-Jacques et al., 1998*). At this stage, the embryonic hair germ has formed, consisting of an epithelial placode and a dermal condensate, whose epithelial-mesenchymal crosstalk is essential for further HF development (*Hardy, 1992*; *Schmidt-Ullrich and Paus, 2005*).

De novo HF formation in adult skin has been observed in combination with wounding in rabbits, mice, and humans (*Breedis, 1954*; *Ito et al., 2007*; *Kligman and Strauss, 1956*; *Lim et al., 2018*), and in unwounded skin as a response to forced epithelial Wnt/β-catenin signaling in mice (*Gat et al., 1998*; *Lo Celso et al., 2004*). Two decades after the initial discovery that Wnt/β-catenin-pathway activation results in ectopic HFs (*Gat et al., 1998*), it is still the only known approach to induce de novo HFs in adult unwounded skin. Accordingly, it remains a major clinical challenge to generate replacement skin with hair, urging the search for new ways to achieve HF formation in adult skin.

In Sonic hedgehog knock out (Shh-/-) mice, HF morphogenesis does not progress beyond the hair germ stage (*St-Jacques et al., 1998*). Notably, compared to wild type skin, Shh-/- hair germs have normal levels of Wnt/β-catenin signaling but reduced Hh-target activation in both the placode and the dermal condensate (*Chiang et al., 1999*; *St-Jacques et al., 1998*). Despite the well-known importance of Hh signaling for HF morphogenesis during embryonic skin development (*Botchkarev and Paus, 2003*; *Mill et al., 2003*; *St-Jacques et al., 1998*) and in wound-induced HF formation (*Lim et al., 2018*), little is known about its potential for de novo HF induction in adult unwounded skin.

**eLife digest** We are born with all the hair follicles that we will ever have in our life. These structures are maintained by different types of cells (such as keratinocytes and fibroblasts) that work together to create hair. Follicles form in the embryo thanks to complex molecular signals, which include a molecular cascade known as the Hedgehog signaling pathway.

After birth however, these molecular signals are shut down to avoid conflicting messages – inappropriate activation of Hedgehog signaling in adult skin, for instance, leads to tumors. This means that our skin loses the ability to make new hair follicles, and if skin is severely damaged it cannot regrow hair or produce the associated sebaceous glands that keep skin moisturized.

Being able to create new hair follicles in adult skin would be both functionally and aesthetically beneficial for patients in need, for example, burn victims. Overall, it would also help to understand if and how it is possible to reactivate developmental programs after birth.

To investigate this question, Sun, Are et al. triggered Hedgehog signaling in the skin cells of genetically modified mice; this was done either in keratinocytes, in fibroblasts, or in both types of cells. The experiments showed that Hedgehog signaling could produce new hair follicles, but only when activated in keratinocytes and fibroblasts together. The process took several weeks, mirrored normal hair follicle development and resulted in new hair shafts. The follicles grew on both the back of mice, where hair normally occurs, and even in paw areas that are usually hairless.

Not unexpectedly the new hair follicles were accompanied with skin tumors. But, promisingly, treatment with Hedgehog-pathway inhibitor Vismodegib restricted tumor growth while keeping the new follicles intact. This suggests that future work on improving "when and where" Hedgehog signaling is activated may allow the formation of new follicles in adult skin with fewer adverse effects.

Interestingly, already decades ago it has been observed that basal cell carcinoma (BCC) morphologically mimics HF development until the hair germ stage. BCC develops upon supra-physiological Hh-pathway activation in epithelial cells, with the most prevalent mutations in the inhibitory Hh-receptor gene *Ptch1*. Like HF formation, Hh-driven BCC is characterized by active Wnt/b-catenin signaling (*Yang et al., 2008*), and early epithelial BCC buds express typical HF-lineage markers even if they don't originate from HFs (*Kasper et al., 2011*; *Yang et al., 2008*; *Youssef et al., 2012*). Morphologically and molecularly, these early BCC buds are thus very similar to embryonic hair germs with one major difference: BCC buds lack a dermal condensate which serves as a focal point for dermal Hh signaling and is required for HF morphogenesis (*Yang et al., 2008*).

Based on these observations, we hypothesized that HF development past the hair germ stage in BCC requires Hh signaling at increased levels in both epithelial and stromal cells. To address this hypothesis, we used mouse models to induce supra-physiological Hh signaling via *Ptch1* deletion in the epithelium and stroma, which indeed led to the induction of new HFs in adult unwounded skin.

## Results

### Activated Hh signaling in *Gli1*-expressing cells induced HF-like structures in touch domes

To test whether activation of epithelial and stromal Hedgehog signaling would result in HF formation, we focused on the touch domes (TDs), which are touch-receptive structures located within the interfollicular epidermis (IFE) (*Figure 1A,B*). We reasoned that in adult unwounded skin, TDs were the most likely places to achieve experimentally induced de novo HFs because TDs are hotspots for BCC formation (*Peterson et al., 2015*), which resembles HF development (*Yang et al., 2008*).

We analyzed and compared the TDs of two mouse models with induced supra-physiological Hh-pathway levels in epithelial cells (Lgr6 mouse model), or combined epithelial and stromal cells of the TD (Gli1 mouse model). Specifically, we used *Lgr6-EGFP-IRES-Cre$^{ERT2}$;R26$^{tdTomato}$;Ptch1$^{fl/fl}$* mice (hereafter: Lgr6$^{creERT2}$;R26$^{Tom}$;Ptch1$^{fl/fl}$) and *Gli1-Cre$^{ERT2}$;R26$^{tdTomato}$;Ptch1$^{fl/fl}$* mice (hereafter: Gli1-$^{creERT2}$;R26$^{Tom}$;Ptch1$^{fl/fl}$). TD areas were identified by the presence of K8+ Merkel cells, and their palisading epithelial cell morphology. Tamoxifen was administered at 8 weeks of age, resulting in the

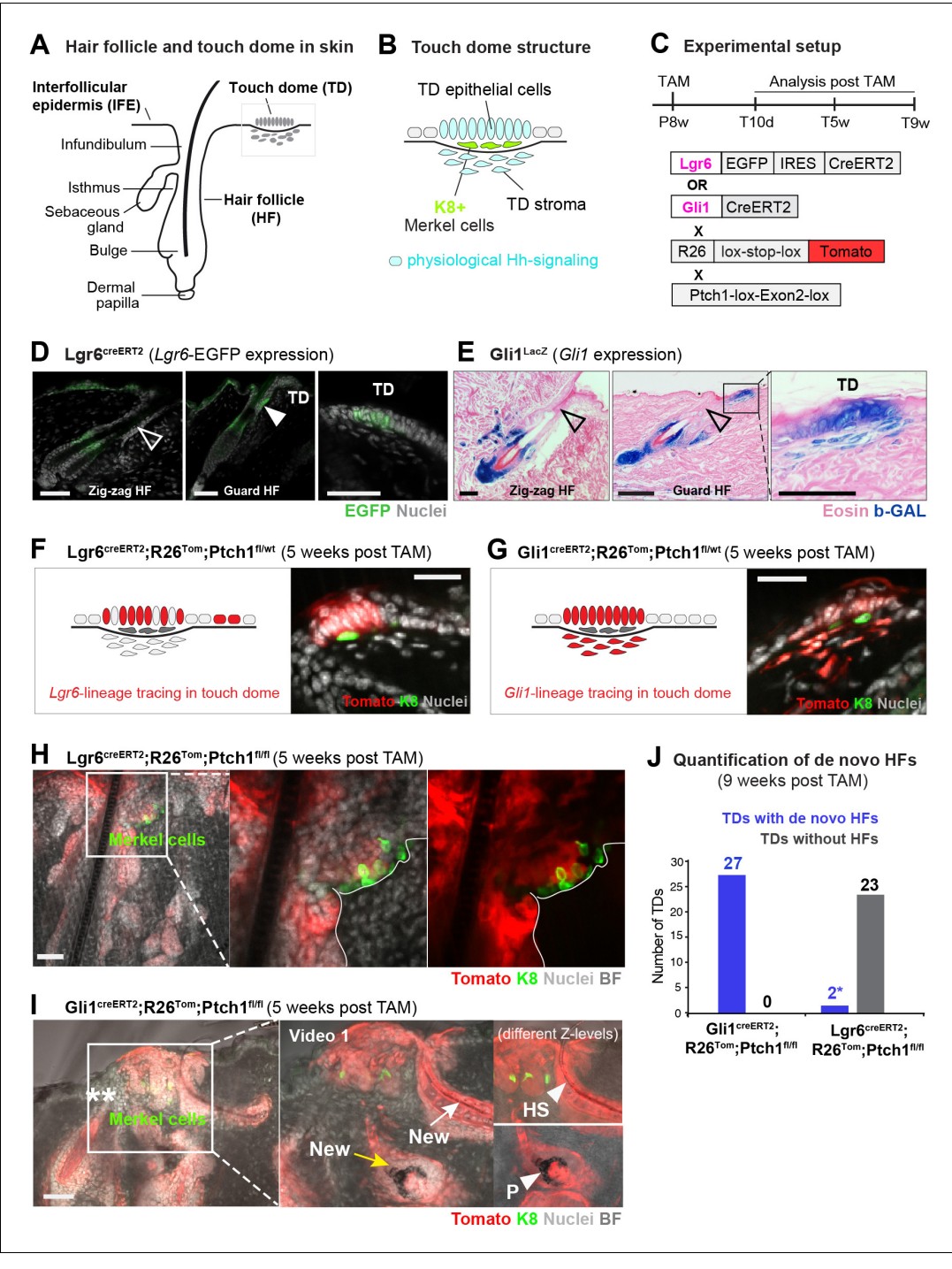

**Figure 1.** Formation of hair follicle (HF)-like structures in touch domes (TDs) of the Gli1 mouse model. (**A–B**) Illustrative cartoon of HF and TD structures in wild type skin. Physiological Hh signaling is present in both TD epithelium and TD stroma. The presence of K8+ Merkel cells is characteristic for TDs. (**C**) Schematic representation of the experimental timeline and the Lgr6$^{creERT2}$;R26$^{Tom}$;Ptch1$^{fl/fl}$ and Gli1$^{creERT2}$;R26$^{Tom}$;Ptch1$^{fl/fl}$ mouse models. (**D–E**) *Lgr6*$^{EGFP}$ and *Gli1*$^{LacZ}$ expression in dorsal telogen skin. Filled arrowhead: indicates *Lgr6*-expression in the HF infundibulum. Empty arrowheads: indicate lack of *Lgr6*- or *Gli1*-expression in HF infundibula (n = 3 mice per genotype). (**F–I**) Mice were treated with tamoxifen (TAM) at 8 weeks of age and dorsal skin was analyzed 5 weeks post TAM. For each genotype ≥3 mice and numerous TDs were analyzed (***Supplementary file 1***). (**F**) Illustrative cartoon and experimental Tomato-tracing of *Lgr6*-expressing cells. TDs of Lgr6$^{creERT2}$;R26$^{Tom}$;Ptch1$^{fl/wt}$ control skin were phenotypically normal. (**G**) Illustrative cartoon and experimental Tomato-tracing of *Gli1*-expressing cells. TDs of Gli1$^{creERT2}$;R26$^{Tom}$;Ptch1$^{fl/wt}$ control skin were phenotypically normal. (**H**) Tomato-traced TD of Lgr6$^{creERT2}$;

*Figure 1 continued on next page*

*Figure 1 continued*

R26$^{Tom}$;Ptch1$^{fl/fl}$ skin displaying basal cell carcinoma (BCC)-like tumor growth. (I) Tomato-traced TD of Gli1$^{creERT2}$; R26$^{Tom}$;Ptch1$^{fl/fl}$ skin with BCC-like tumor growth and several de novo HFs. Asterisks: mark a non-traced infundibulum in the pre-existing Guard HF adjacent to the TD. Inset: different z-levels better depicting specific de novo HF structures. A video containing all z-levels is provided (*Figure 1—video 1*). Arrows: de novo HFs with continuous tracing into the IFE. Yellow arrow: anagen bulb of a de novo HF. Arrowheads: hair shaft (HS) or pigment (P). (J) Quantification of de novo HFs in the TDs of Gli1$^{creERT2}$;R26$^{Tom}$;Ptch1$^{fl/fl}$ and Lgr6$^{creERT2}$;R26$^{Tom}$; Ptch1$^{fl/fl}$ mice (9 weeks post TAM; n = 3 mice for each genotype). Asterisk: the Lgr6 mouse model cannot inform on de novo HFs trough lineage tracing; based on morphology we observed in two TDs a single potentially new hair shaft, respectively. TD: touch dome. HS: hair shaft. P: pigment. TAM: tamoxifen. BF: bright field. Scale bars: 50 μm (D–I).

The online version of this article includes the following video and figure supplement(s) for figure 1:

**Figure supplement 1.** Panorama to *Figure 1H*.
**Figure supplement 2.** Phenotype and lineage-tracing pattern in Lgr6$^{creERT2}$;R26$^{Tom}$;Ptch1$^{fl/wt}$ and Lgr6$^{creERT2}$; R26$^{Tom}$;Ptch1$^{fl/fl}$ skin.
**Figure supplement 3.** Panorama to *Figure 1I*.
**Figure 1—video 1.** Video containing all recorded z-levels of the touch dome area presented in *Figure 1I*.
https://elifesciences.org/articles/46756#fig1video1

---

constitutive activation of Hh signaling via homozygous inactivation of *Ptch1* and simultaneous Tomato-tracing of *Lgr6*- or *Gli1*-expressing cells (*Figure 1C*). First, we confirmed that *Lgr6* expression, and consequently Tomato-tracing, were in the TD restricted to epithelial cells (*Figure 1D,F*), and *Gli1* expression and Tomato-tracing were present in both epithelial and stromal TD cells (*Figure 1E,G*). Next, we analyzed the phenotypes of both Lgr6 and Gli1 mouse models 5 weeks post tamoxifen, a sufficiently long time to allow possible de novo HFs to form (*Rendl et al., 2005*). Homozygous *Ptch1* inactivation in *Lgr6*-expressing cells resulted as expected in BCC-like lesions in HFs, IFE and TDs (*Figure 1H*, *Figure 1—figure supplement 1*; *Peterson et al., 2015*). Strikingly, however, homozygous *Ptch1* inactivation in *Gli1*-expressing cells resulted in addition to BCC-like lesions in HFs and TDs, in the formation of structures in TDs that resembled de novo HFs (*Figure 1I*, *Figure 1—figure supplement 3*). These structures had the appearance of typical concentric anagen HF layers and a pigmented hair bulb or mature hair shaft (*Figure 1I*, *Figure 1—video 1*). Importantly, by 9 weeks post tamoxifen, these HF-like structures occurred in every single Gli1$^{creERT2}$; R26$^{Tom}$;Ptch1$^{fl/fl}$ TD examined (27/27) but extremely rarely in TDs of Lgr6$^{creERT2}$;R26$^{Tom}$;Ptch1$^{fl/fl}$ mice (2/25; please also see Discussion) (*Figure 1J*). No such HF-like structures were observed in TDs of wild type mice (with physiological Hh signaling in epithelial and stromal cells), upon heterozygous *Ptch1* inactivation (Gli1$^{creERT2}$;R26$^{Tom}$;Ptch1$^{fl/wt}$ and Lgr6$^{creERT2}$;R26$^{Tom}$;Ptch1$^{fl/wt}$) (*Figure 1F,G*), or in non-tamoxifen controls (Gli1$^{creERT2}$;R26$^{Tom}$;Ptch1$^{fl/fl}$ and Lgr6$^{creERT2}$;R26$^{Tom}$;Ptch1$^{fl/fl}$) (*Supplementary file 1*). Therefore, induction of supra-physiological Hh signaling in epithelial and stromal cells (Gli1 model) but not epithelial cells alone (Lgr6 model) was sufficient to induce HF-like structures in TDs of adult mouse skin.

## Characterization of de novo HFs in touch domes

Next we investigated whether the observed structures were functional HFs, and indeed de novo induced. Thus, we stained the skin of Gli1$^{creERT2}$;R26$^{Tom}$;Ptch1$^{fl/fl}$ mice for Keratin 71 (K71) and Keratin 6 (K6) (*Figure 2A*), which mark specific layers of the anagen HF (*Yang et al., 2017*). These Keratin-staining patterns were very similar to those of hair-cycle stage-matched wild type anagen HFs (*Figure 2B*). Also, the presence, and specific pattern of hair pigment in these structures were typical for anagen HFs, in the shown picture matching the anagen III hair-cycle stage (*Figure 2A,B*), further supporting that the observed structures are indeed HFs and actively growing. At 5 weeks post tamoxifen all de novo HFs were in different stages of anagen (*Figures 1I* and *2A*) and by 9 weeks post tamoxifen the majority of de novo HFs were in telogen (*Figure 2C*, *Figure 2—figure supplement 2*). This demonstrates that de novo HFs enter the hair cycle after their first anagen (*Paus and Cotsarelis, 1999*).

To verify that these HFs were de novo induced, thorough examination of the Tomato lineage tracing pattern was of key importance. The *Gli1*-positive HF and TD populations self-renew

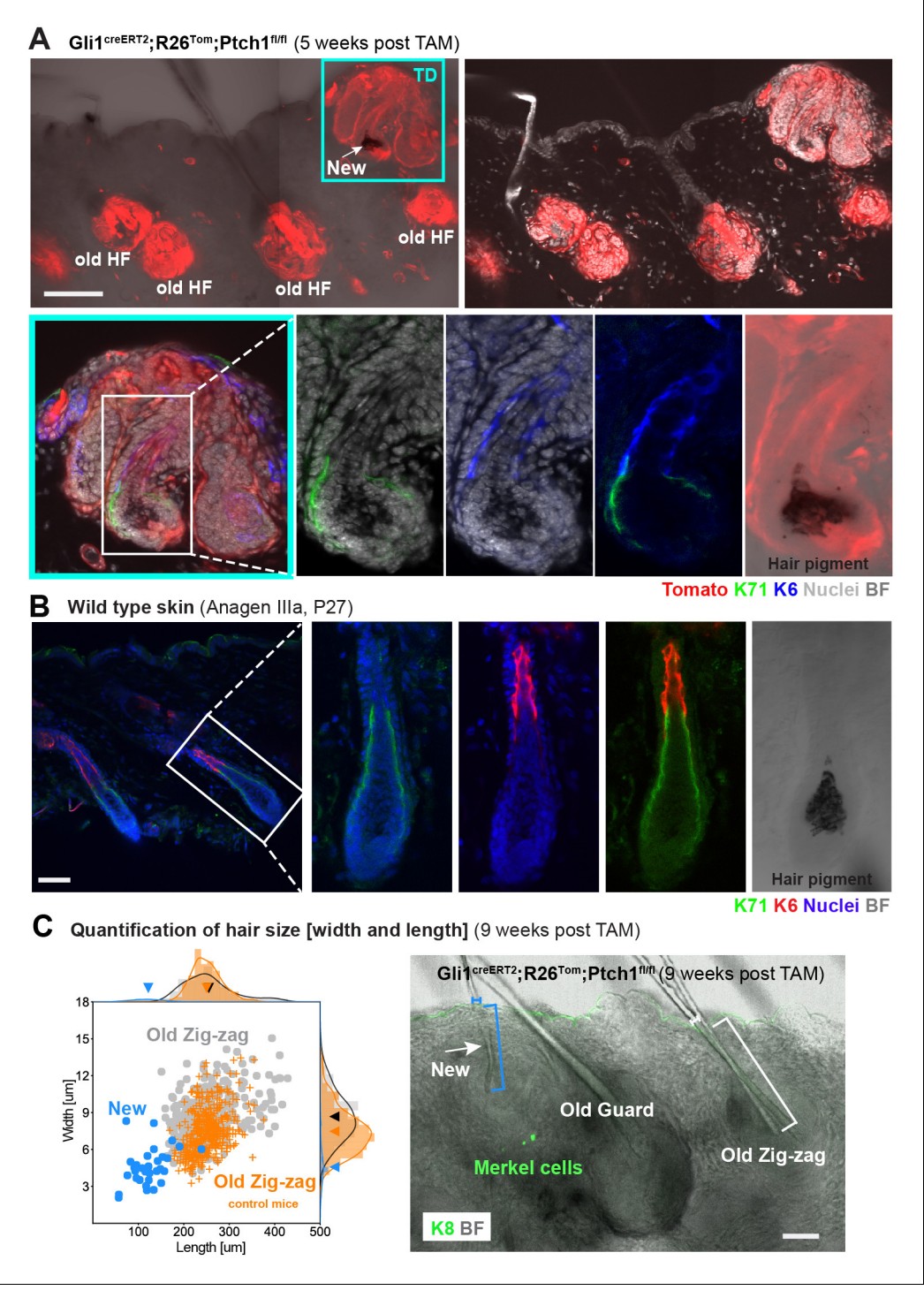

**Figure 2.** Characterization of de novo hair follicles (HFs) in Gli1<sup>creERT2</sup>;R26<sup>Tom</sup>;Ptch1<sup>fl/fl</sup> touch domes (TDs). (A) Gli1<sup>creERT2</sup>;R26<sup>Tom</sup>;Ptch1<sup>fl/fl</sup> mice were treated with tamoxifen (TAM) at 8 weeks and dorsal skin was analyzed 5 weeks post TAM treatment (n = 3 mice). The TD area shows a de novo anagen HF (turquois frame). Additionally, traced pre-existing (old) telogen HFs with basal cell carcinoma (BCC)-like growth are present. Inset (white frame): anagen hair bulb of de novo HF showing K71-positive Henle's layer, K6-positive companion layer, hair pigment and continuous Tomato-tracing from the hair bulb into the TD. (B) Immunofluorescent co-staining of K71 (Henle's layer) and K6 (companion layer) in a wild type HF of a similar hair cycle stage (Anagen IIIa, P27) (n = 2 mice). (C) Quantification of hair size in Gli1<sup>creERT2</sup>;R26<sup>Tom</sup>;Ptch1<sup>fl/fl</sup> and Gli1<sup>creERT2</sup>;R26<sup>Tom</sup>;Ptch1<sup>fl/wt</sup> mice that were treated with TAM at 8 weeks and dorsal skin was analyzed 9 weeks post TAM treatment. Right panel: De novo telogen

*Figure 2 continued on next page*

*Figure 2 continued*

HFs with a thin hair shaft formed in the TDs of Gli1$^{creERT2}$;R26$^{Tom}$;Ptch1$^{fl/fl}$ skin (white arrow). For the quantification, we analyzed hair shafts of de novo HFs from Gli1$^{creERT2}$;R26$^{Tom}$;Ptch1$^{fl/fl}$ mice (blue bracket), old/pre-existing Zig-zag HFs from the same mice (white bracket), and Zig-zag HFs from wild-type-phenotype control mice (Gli1$^{creERT2}$; R26$^{Tom}$;Ptch1$^{fl/wt}$) (n = 3 mice for each genotype; 34 de novo, 314 old/pre-existing, and 437 control HFs; *Figure 2—source data 1*). Hair shaft length was measured in telogen stage hair shafts from the hair club to the HF opening (as indicated by the blue and white brackets). Left panel: Each dot represents a hair shaft and the dots are colored according to the HF type (blue for de novo HFs, gray for pre-existing Zig-zag HFs, orange for control Zig-zag HFs). De novo hair shafts were significantly smaller (p-value<1.10$^{-6}$) and thinner (p-value<1.10$^{-6}$) compared to old/pre-existing and control Zig-zag hairs (Mann-Whitney U test). Arrowheads on the x- and y-axis indicate the mean values of hair shaft length and width of de novo, old/pre-existing or control Zig-zag HFs, respectively. TD: touch dome. HF: hair follcile. TAM: tamoxifen. BF: bright field. Scale bars: 100 µm (**A**), 50 µm (**B**, **C**).

The online version of this article includes the following source data and figure supplement(s) for figure 2:

**Source data 1.** Quantification of hair width and length.
**Figure supplement 1.** Lineage-tracing pattern of TD and surrounding area in Gli1$^{creERT2}$;R26$^{Tom}$;Ptch1$^{fl/wt}$ control and Gli1$^{creERT2}$;R26$^{Tom}$;Ptch1$^{fl/fl}$ skin during de novo HF formation.
**Figure supplement 2.** De novo HFs in TDs of Gli1$^{creERT2}$;R26$^{Tom}$;Ptch1$^{fl/fl}$ mice 9 weeks post TAM were in telogen and displayed a HF bulge.

independently with a clear *Gli1*-negative gap in the infundibulum (*Figure 1E*; *Xiao et al., 2015*). Cell crossover only occurs for example in response to full-thickness wounding or TPA treatment (*Brownell et al., 2011*; *Kasper et al., 2011*). Importantly, *Ptch1* deletion did not trigger HF cell migration towards the IFE or TD as we never observed Tomato-tracing spanning from a pre-existing HF – via the infundibulum – to the IFE or TD in any control or Gli1$^{creERT2}$;R26$^{Tom}$;Ptch1$^{fl/fl}$ skin (*Figure 1—figure supplement 3*; *Figure 2—figure supplement 1*). A non-traced infundibulum, leaving a tracing gap between the HF and the IFE/TD, is thus characteristic for pre-existing HFs. In contrast, all de novo HFs displayed continuous Tomato-tracing from the de novo HF to its originating TD, including the infundibulum and all anagen HF lineages (*Figures 1I* and *2A*, *Figure 1—video 1* and *Figure 2—figure supplement 1*). As such a continuous HF-to-TD-tracing pattern can only result from newly formed HFs originating from traced TD-epithelial cells, we unequivocally demonstrated that these HFs were new.

An additional characteristic of de novo HFs was that their hair shafts were considerably shorter and thinner than hair shafts from both pre-existing HFs in the same mice (Gli1$^{creERT2}$;R26$^{Tom}$;Ptch1$^{fl/fl}$) and HFs in control mice with wild type phenotype (Gli1$^{creERT2}$;R26$^{Tom}$;Ptch1$^{fl/wt}$) (*Figure 2C* and *Figure 2—source data 1*). In conclusion, the continuous lineage-tracing from HF to TD, and the characteristic hair shaft measurements, demonstrated that combined epithelial and stromal Hh-pathway activation in the Gli1 mouse model via homozygous *Ptch1* inactivation resulted in de novo HFs, within TDs.

## De novo HF formation recapitulates embryonic HF development

Next we characterized the stages of de novo HF development in TDs equivalent to the embryonic HF developmental stages of HF placode, hair germ, hair peg and mature follicle (*Rendl et al., 2005*). We analyzed Gli1$^{creERT2}$;R26$^{Tom}$;Ptch1$^{fl/fl}$ and control skin, tamoxifen treated at 8 weeks of age, and collected samples 10, 17, 25, 27, 29, 33, 35 and 36 days after tamoxifen treatment. These sampling time points enabled us to map the entire stereotypical time course of de novo HF formation (*Figure 3A–D*, *Supplementary file 1*). In some TDs, the HF-placode stage could be already detected 10 days post tamoxifen administration, and was accompanied by the appearance of a dermal condensate (*Figure 3A*). The dermal condensate and dermal papillae were delineated by denser Tomato-tracing of stromal cells compared to adjacent traced K5+ epithelial cells. Note the lack of continuous pre-existing HF-to-TD tracing at this early developmental stage (*Figure 3A,E*), again supporting that the fully traced HFs (*Figures 1I*, *2A* and *3B–D*) were newly formed and induced from traced TD cells. The hair germ and hair peg stages were mostly detected between 27 and 29 days post tamoxifen administration (*Figure 3B,C*), and mature HFs, with a clearly visible developing hair shaft, emerged mainly at or after 33 days post tamoxifen administration (*Figure 3D,D',F*).

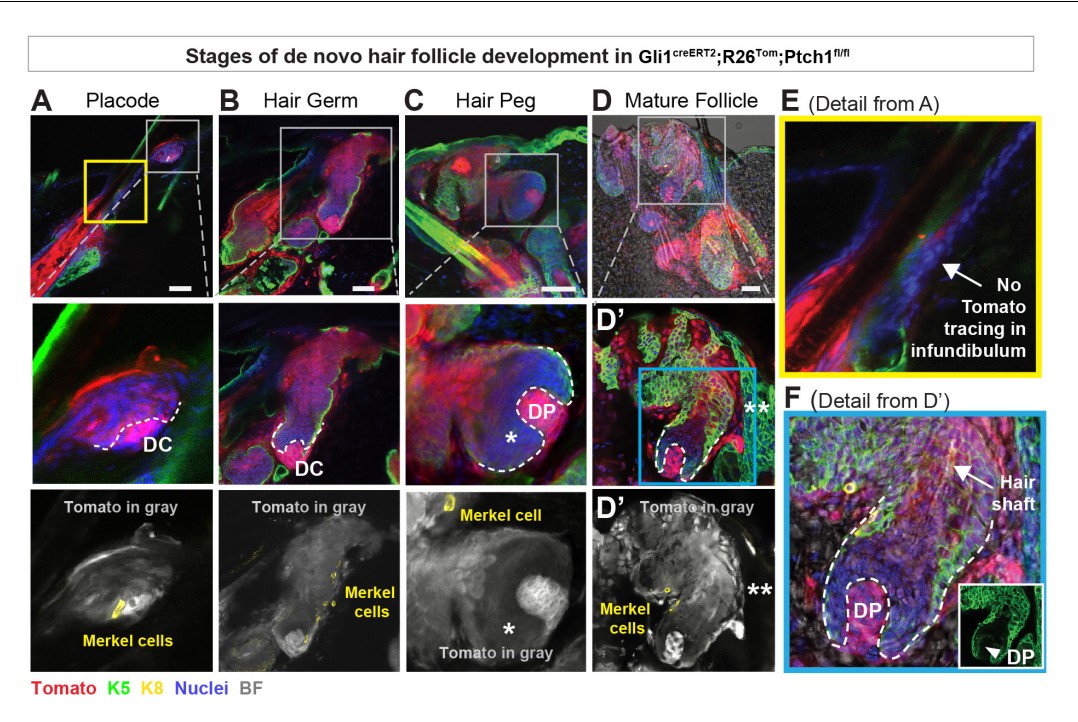

**Figure 3.** Developmental stages of de novo hair follicles (HFs) in Gli1^creERT2;R26^Tom;Ptch1^fl/fl touch domes (TDs). (A–F) Gli1^creERT2;R26^Tom;Ptch1^fl/fl mice were treated with tamoxifen (TAM) at 8 weeks and dorsal skin was analyzed 10–36 days post TAM to characterize de novo HFs originating from TDs in the following developmental stages (n = 6 mice): early placode (A), hair germ (B), hair peg (C), and mature follicle (D). These stages recapitulate embryonic HF development (*Rendl et al., 2005*). Dermal condensates (A, B) or dermal papillae (C, D) are clearly visible, and the de novo HFs – both early and mature – are continuously traced into the TD. Note: HF-matrix cells are Tomato-traced with reduced intensity (* in C), while the infundibulum of pre-existing HFs was not Tomato-traced (** in D' and arrow in E). Furthermore, mature HFs contain a hair shaft (arrow in F). Inset in F shows K5-positive staining of the HF epithelial cells and a K5-negative dermal papilla. DC: dermal condensate. DP: dermal papilla. BF: bright field. Scale bars: 50 µm (A-D).

Interestingly, almost all dermal condensates and dermal papillae of de novo forming HFs were fully Tomato traced (*Figure 3A–D,F*), suggesting that either continuous high levels of Hh signaling in stromal cells (Ptch1^fl/fl) are necessary for all stages of de novo HF development, or that stromal cells with constitutive Hh-pathway activation outcompete dermal condensate and dermal papilla cells with lower Hh-signaling levels. Importantly, Syndecan-1 (SDC1) expression that marks early dermal condensates in wild type embryonic skin (*Figure 4A*; *Richardson et al., 2009*), could already be detected 10 days post tamoxifen in dermal condensates of Gli1^creERT2;R26^Tom;Ptch1^fl/fl skin (*Figure 4B*), and was fully established in the dermal papilla at the hair germ stage (*Figure 4C*).

Finally, to demonstrate that these de novo HFs do indeed have active Hh signaling, we stained for *Gli1* mRNA expression; as a reporter of canonical Hh-pathway activity. The placodes (10 days post tamoxifen) as well as intermediate and mature developmental stages (5 weeks post tamoxifen) expressed *Gli1* mRNA and hence have active Hh/Gli signaling (*Figure 4F–G* and *Figure 4—figure supplement 1C–D*); which is in line with normal embryonic HF development (*Figure 4D* and *Figure 4—figure supplement 1A*; *Chiang et al., 1999*; *St-Jacques et al., 1998*). Wild type TD epithelial and stromal cells expressed *Gli1* mRNA as expected (*Figure 4*E and *Figure 4—figure supplement 1B*).

This *Gli1* RNA-FISH combined with antibody staining for Tomato-lineage tracing also confirmed that Tomato-tracing (cells with *Ptch1* deletion) and *Gli1* expression were highly correlated, as expected (*Figure 4F–G* and *Figure 4—figure supplement 1C–D*). We conclude that de novo HF formation in TDs of adult Gli1^creERT2;R26^Tom;Ptch1^fl/fl skin recapitulates the hallmarks of HF development during embryogenesis.

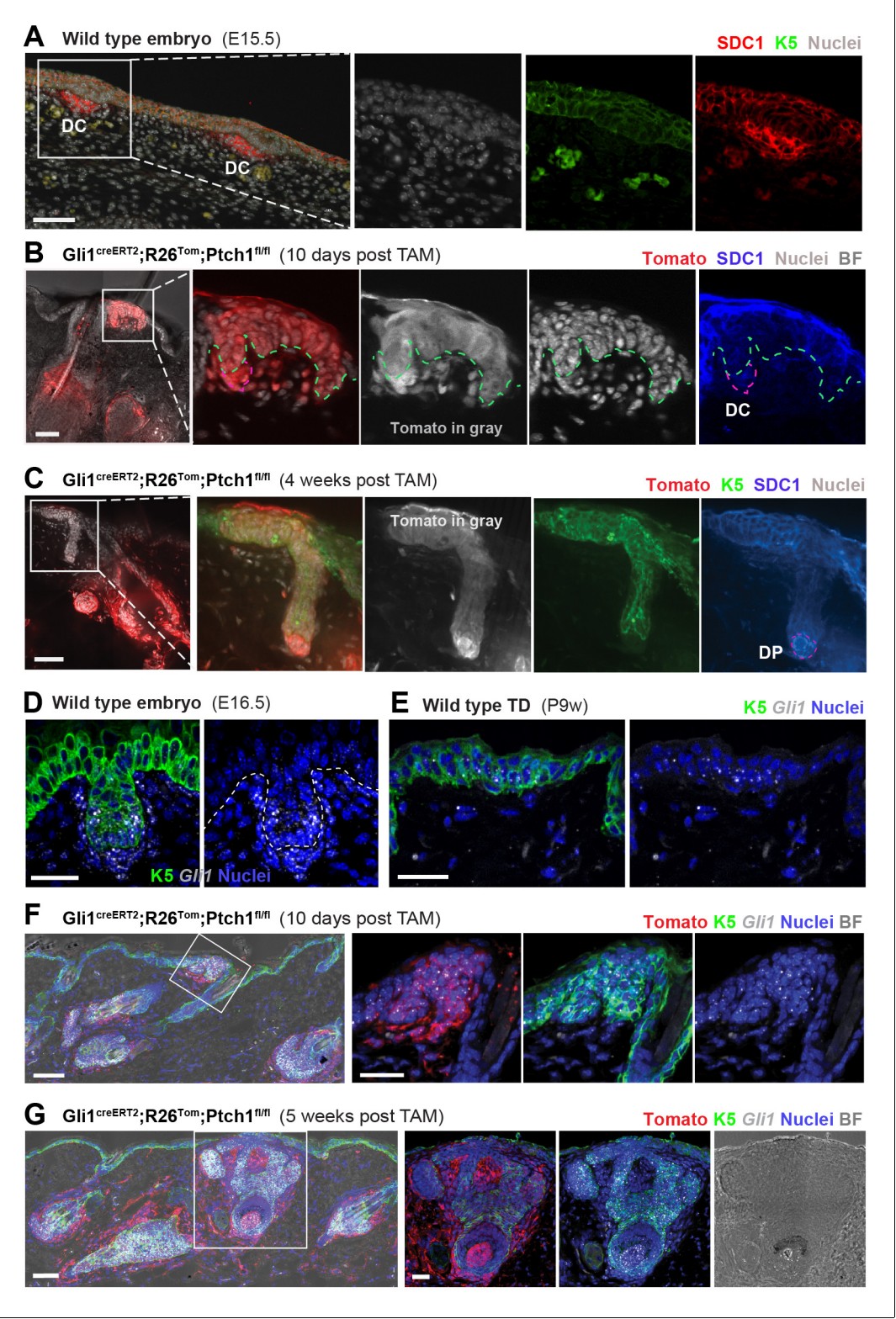

**Figure 4.** SDC1 protein and *Gli1* mRNA expression in developing hair follicles (HFs). (**A**) In wild type embryonic skin (E15.5), Syndecan-1 (SDC1) staining highlights dermal condensates (DCs) (n = 2 mice). (**B–C**) Gli1^creERT2; R26^Tom;Ptch1^fl/fl mice were treated with tamoxifen (TAM) at 8 weeks and dorsal skin was analyzed for SDC1 expression in the newly formed HF buds (n = 3 mice). (**B**) Early placode stage (image from a TD 10 days post TAM) displaying faint SDC1 staining in dermal condensate cells. Comparable to epithelium of wild type embryonic HF

*Figure 4 continued on next page*

*Figure 4 continued*

buds, early de novo placodes also express some SDC1. (**C**) Positive SDC1 staining of dermal papilla (image from a TD 4 weeks post TAM). (**D–E**) *Gli1* RNA-FISH. Both the epithelial placode and dermal condensate have active Hh/Gli signaling in wild type embryonic skin (E16.5) (**D**) and at lower levels in wild type TDs of adult skin (**E**). (**F–G**) Gli1$^{creERT2}$;R26$^{Tom}$;Ptch1$^{fl/fl}$ mice were treated with TAM at 8 weeks and dorsal skin was analyzed for *Gli1* mRNA expression in the placode stage (10 days post TAM) as well as in an intermediate developmental stage (5 weeks post TAM). Active canonical Hh/Gli signaling was present in all analyzed de novo HF stages in epithelial and dermal papilla cells. Note that Tomato-tracing was visualized using an RFP-antibody. Green and white dashed lines: epithelial-stromal border. Purple dashed line: outlines the dermal condensate. DC: dermal condensate. DP: dermal papilla. TD: touch dome. BF: bright field. For RNA-FISH stainings, n = 2 mice (**D, E and G**) and n = 1 mouse (**F**). Scale bars: 25 µm (**D–G**), 50 µm (**A-C, F-G** panoramas).

The online version of this article includes the following figure supplement(s) for figure 4:

**Figure supplement 1.** *Gli1* mRNA staining reveals cells with active canonical Hh/Gli signaling during de novo HF development in the TDs of Gli1$^{creERT2}$;R26$^{Tom}$;Ptch1$^{fl/fl}$ mice.

## De novo HFs persist while BCC-like tumors diminish upon vismodegib treatment

De novo HF formation in the Gli1$^{creERT2}$; R26$^{Tom}$; Ptch1$^{fl/fl}$ mouse model is accompanied with BCC growth; that is BCC-like lesions appear in pre-existing HF as well as in TDs (*Figure 1—figure supplement 3*). In TDs, the clearly identifiable de novo HFs develop alongside epithelial tumor growth which is characterized by palisading cells and lack of HF-like structures. It has been shown previously that BCC-like lesions in dorsal skin dramatically shrink within seven days upon vismodegib treatment (*Eberl et al., 2018*). Vismodegib is a Hh-pathway inhibitor acting at the level of Smoothened (Smo), and using the optimized treatment scheme from *Eberl et al. (2018)* we tested whether established de novo HFs would persist or would diminish as the BCC-tumor-growth area does. We treated Gli1$^{creERT2}$;R26$^{Tom}$;Ptch1$^{fl/fl}$ mice with tamoxifen at 8 weeks. Five to seven weeks after tamoxifen treatment when de novo HFs were clearly established in TDs, we took a dorsal biopsy prior to vismodegib treatment (untreated biopsy), and then treated the mice daily with vismodegib (50 mg/kg body weight i.p.) for seven days (*Figure 5A*).

Reassuringly, in pre-existing HFs we found considerable reduction of tumor size (*Figure 5B*) as well as absence of Ki67 staining in Tomato-traced areas when comparing 7 day vismodegib treated samples with the untreated biopsies of the same mice (*Figure 5C*). This reduction in tumor size demonstrated that vismodegib treatment worked as expected. In TDs, the tumor areas also dramatically diminished in size and Ki67 staining, however the de novo HFs persisted (*Figure 5D* and *Figure 5—figure supplement 1*). In conclusion, these experiments confirmed that de novo HFs indeed represent HFs that are independent of tumor structures as they persist upon vismodegib treatment when the surrounding BCC-tumor-growth areas are nearly gone.

## Stromal Hh-pathway activation alone is not sufficient to induce HF neogenesis

De novo HFs were induced by strong activation of Hh signaling (Ptch1$^{fl/fl}$) in epithelial and adjacent stromal cells and persisted upon vismodegib treatment when the surrounding BCC-growth areas were nearly gone. As BCC growth merely depends on epithelial Hh-pathway activation, it was tempting to test whether homozygous inactivation of *Ptch1* exclusively in stromal cells would be sufficient to induce de novo HFs without tumor development. To that end, we generated the Col1a2 mouse model (Col1a2-Cre$^{ER}$;R26$^{tdTomato}$;Ptch1$^{fl/fl}$, hereafter: Col1a2$^{creER}$;R26$^{Tom}$;Ptch1$^{fl/fl}$) (*Figure 6A,B*), which drives supra-physiological Hh signaling in the stromal compartment only, via the collagen type I alpha two chain promoter. Non-tamoxifen controls (Col1a2$^{creER}$;R26$^{Tom}$;Ptch1$^{fl/fl}$ mice) showed some tracing in the skin stroma, which however did not result in an adverse skin phenotype except for earlier anagen entry (*Figure 6—figure supplement 1*). Administration of tamoxifen at 8 weeks of age resulted in substantial Tomato-tracing that was restricted to the stromal skin compartment (*Figure 6—figure supplement 2*), and importantly, the stromal cells of the TD were also traced (*Figure 6D*). Homozygous *Ptch1* inactivation in *Col1a2*-expressing cells resulted in increased stromal cell density in TDs (*Figure 6E*), but did not result in de novo HF formation (*Figure 6C,E*), nor did stromal cells stain positive for SDC1 even 9 weeks after tamoxifen treatment

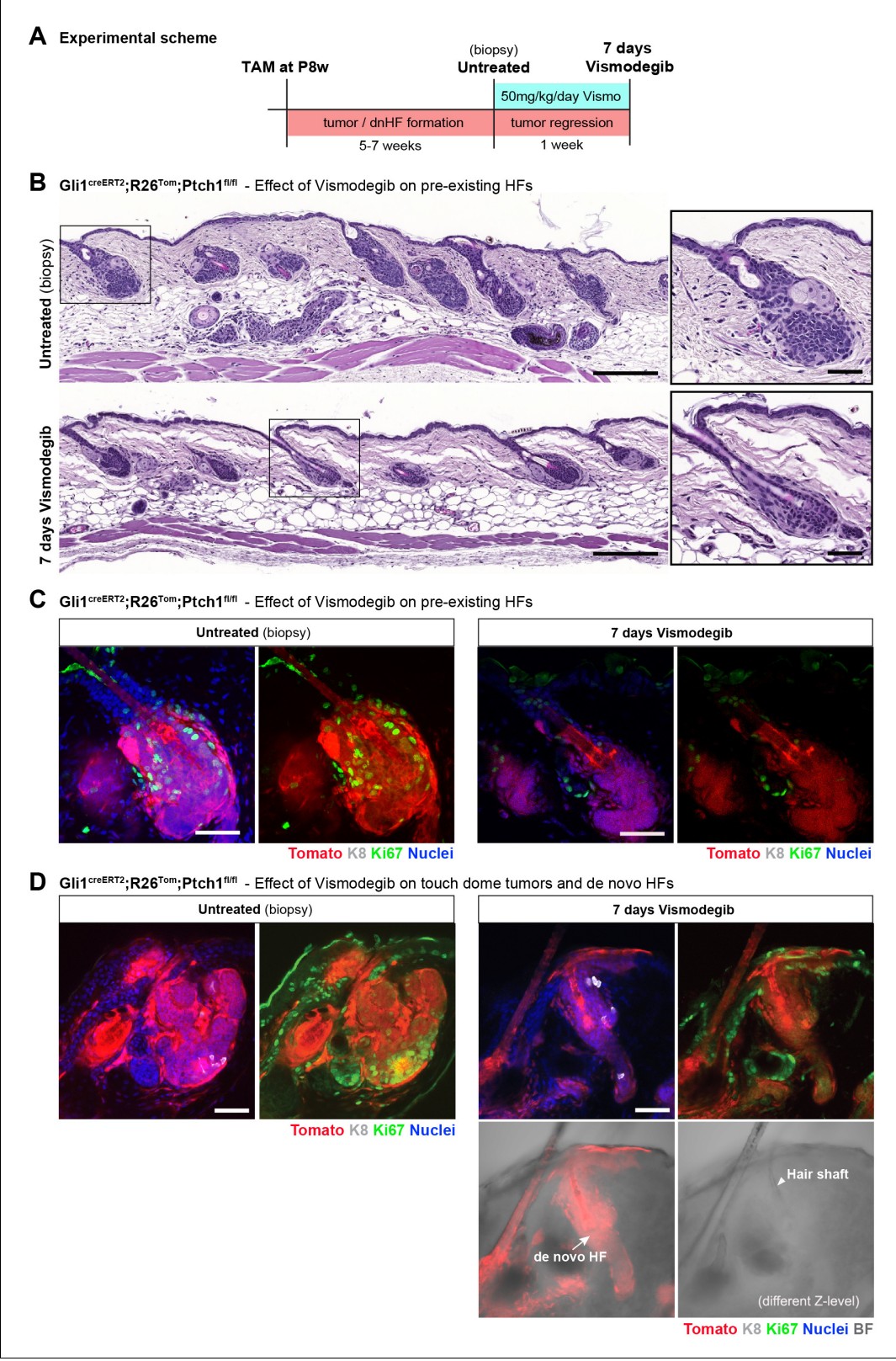

**Figure 5.** Established de novo hair follicles (HFs) in touch domes (TDs) persist upon short-term vismodegib treatment. (A) Experimental scheme of vismodegib treatment. Gli1$^{creERT2}$;R26$^{Tom}$;Ptch1$^{fl/fl}$ mice were treated with tamoxifen (TAM) at 8 weeks of age. Five to seven weeks post TAM treatment, when de novo HFs were clearly established in the TDs, vismodegib was given daily for a week and dorsal skin was analyzed (n = 3 mice for

*Figure 5 continued on next page*

*Figure 5 continued*

Gli1$^{creERT2}$;R26$^{Tom}$;Ptch1$^{fl/fl}$, n = 1 mouse for Gli1$^{creERT2}$;Ptch1$^{fl/fl}$). (**B**) Hematoxylin and eosin stainings showing that basal cell carcinoma (BCC)-like lesions were considerably reduced in response to a week of daily vismodegib treatment at a dose of 50 mg/kg body weight. (**C**) Tumor-cell proliferation in bulge area of pre-existing HFs assessed by Ki67 immunostaining. Bulge areas of untreated control biopsies showed high proliferation, which was almost entirely stalled in 7-day vismodegib samples; in the HF, only the sebaceous glands retained Ki67 expression. (**D**) BCC-like lesions were present in the TDs of dorsal biopsies taken prior to vismodegib treatment (left panel). In response to vismodegib, the tumor-growth area was considerably reduced while de novo HFs presisted (arrow). The de novo HFs are fully Tomato-traced and have a clearly visible hair shaft (arrowhead). HF: hair follicle. K8: marking TD area. Ki67: marking proliferating cells. TAM: tamoxifen. BF: bright field. Scale bars: 200 µm (**B** panaroma), 50 µm (**B** inset, **C**, **D**).

The online version of this article includes the following figure supplement(s) for figure 5:

**Figure supplement 1.** Effect of vismodegib on BCC-like lesions and de novo HFs in TDs.

(*Figure 6F*). We also detected fully traced and highly condensed dermal clusters of cells (resembling dermal condensates) in TD-adjacent infundibula and underneath the regular IFE (*Figure 6G*, *Figure 6—figure supplement 2B*), which did not result in de novo HF induction and the dermal cell condensations were entirely negative for SDC1 expression (*Figure 6G*). We conclude that stromal activation of Hh signaling (Ptch1$^{fl/fl}$) leads to increased stromal cell density and formation of cell condensates, however it is not sufficient to induce HF neogenesis in TDs nor elsewhere in skin without adjacent epithelial Hh-pathway activation.

## Only TDs of the Gli1 mouse model are competent for HF initiation

To directly compare all three different mouse models (Lgr6$^{creERT2}$/, Gli1$^{creERT2}$/, Col1a2$^{creER}$;R26$^{Tom}$; Ptch1$^{fl/fl}$) in their competence to initiate de novo HFs, we analyzed TDs 10 days post tamoxifen administration, the time point when epithelial proliferation became evident via, for example, increased BrdU incorporation in hair-forming TDs before appearance of morphological hair germ formation (*Figure 7*, *Figure 7—figure supplement 1*). We investigated the stromal TD compartment using the alkaline phosphatase (AP) assay and SDC1 staining, which are both characteristic for HF-inducing dermal condensates (*Ito et al., 2007*; *Richardson et al., 2009*). Positive staining for both alkaline phosphatase (ALPL) and SDC1 expression was observed in the Gli1$^{creERT2}$;R26$^{Tom}$;Ptch1$^{fl/fl}$ mice, but not in Gli1$^{creERT2}$;R26$^{Tom}$;Ptch1$^{fl/wt}$, Lgr6$^{creERT2}$;R26$^{Tom}$;Ptch1$^{fl/fl}$, or Col1a2$^{creER}$;R26$^{Tom}$; Ptch1$^{fl/fl}$ mice (*Figure 7A,B*). The epithelial TD compartment could not be stained for a comparable marker of early HF induction, as early BCC buds express typical HF-lineage markers. Indeed, we and others have not found a single mRNA/protein stain that would distinguish HF epithelial placode from BCC formation (*Kasper et al., 2011*; *Yang et al., 2008*; *Youssef et al., 2012*). Altogether, staining for early signs of HF formation demonstrated that only the Gli1 (Ptch1$^{fl/fl}$) model bears TDs that are competent for de novo HF formation.

## Cells outside TD niches can give rise to de novo HFs

We established that de novo HF formation required close apposition of epithelial and stromal Hh signaling (Ptch1$^{fl/fl}$), and furthermore how to identify de novo HFs based on their morphology and continuous lineage tracing. We next asked whether de novo HFs could also form from non-TD areas with comparable adjacent epithelial-stromal Hh-pathway activation.

In addition to TDs, the HF isthmus also harbors adjacent epithelial and stromal *Gli1*-Tomato traced cells; the latter evident through Tomato/PDGFRb co-staining of stromal cells (*Figure 8A,B*). Thus, we re-examined the HF isthmus areas of Gli1$^{creERT2}$;R26$^{Tom}$;Ptch1$^{fl/fl}$ skin for potential de novo HFs. Indeed, at low frequency, we observed de novo HFs in the isthmus of pre-existing HFs that were most likely newly formed (*Figure 8C,D*). Although it was not possible to unequivocally determine de novo formation through continuous Tomato-tracing of the infundibulum (as these de novo HFs seem to merge directly into the isthmus area of pre-existing HFs), based on their morphology, positioning and the co-occurrence of four hair shafts instead of normally three (in dorsal skin of 17 week-old mice), these HFs likely formed newly from the isthmus of pre-existing HFs (*Figure 8C*). Such HFs have not been observed in phenotypically normal control skin (Gli1$^{creERT2}$;R26$^{Tom}$;Ptch1$^{fl/wt}$ and Lgr6$^{creERT2}$;R26$^{Tom}$;Ptch1$^{fl/wt}$) or in skin with HF tumors (Lgr6$^{creERT2}$;R26$^{Tom}$;Ptch1$^{fl/fl}$)

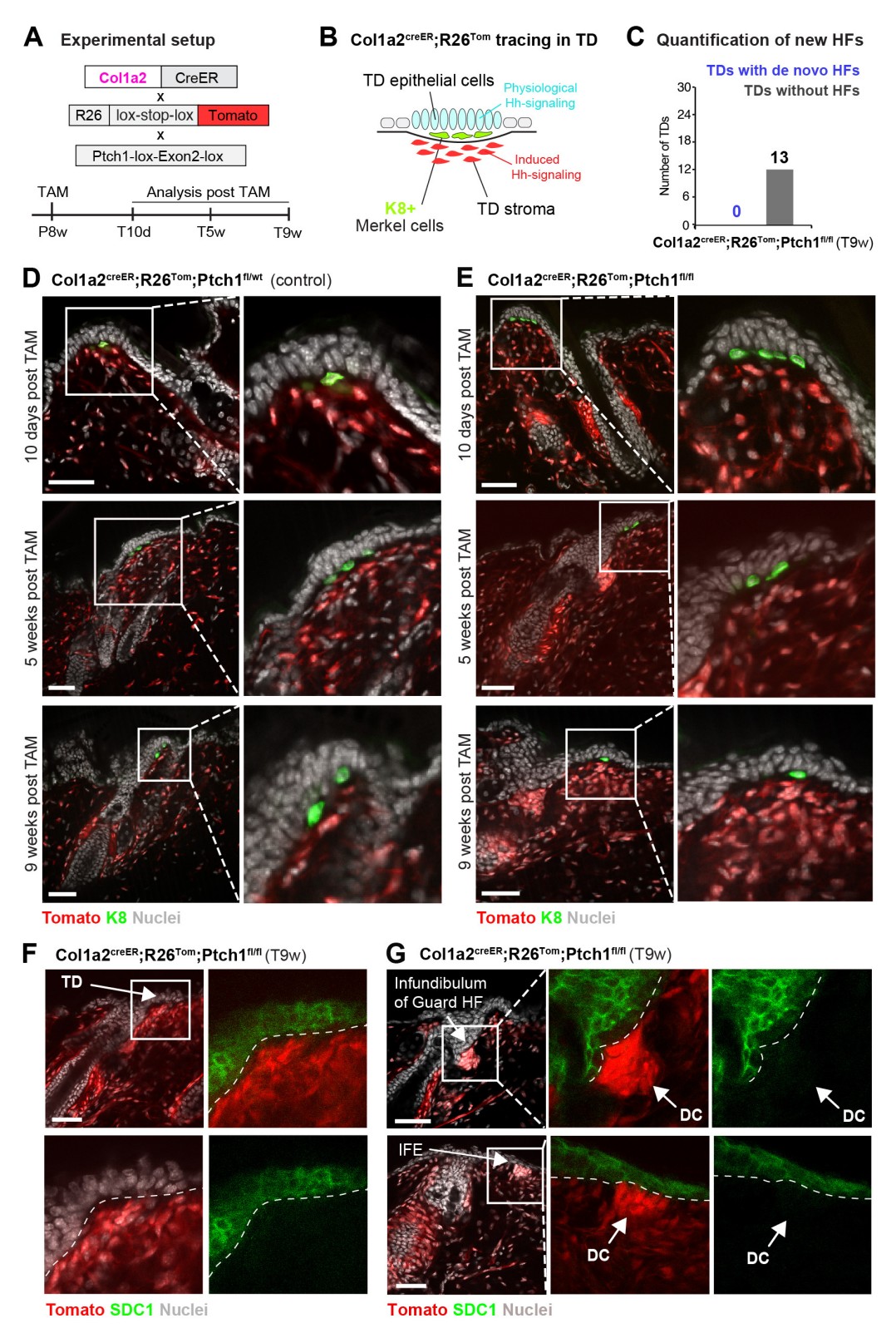

**Figure 6.** Stromal Hh pathway activation alone is not sufficient to induce hair follicle (HF) neogenesis in the touch domes (TDs) of Col1a2$^{creER}$;R26$^{Tom}$;Ptch1$^{fl/fl}$ skin. (**A**) Schematic representation of the Col1a2$^{creER}$;R26$^{Tom}$;Ptch1$^{fl/fl}$ mouse model and the experimental timeline. (**B**) Illustrative cartoon of Tomato-tracing of *Col1a2*-expressing cells and Hh-signaling levels in TD epithelium and TD stroma in this mouse model. (**C**) Quantification of de novo HFs in

*Figure 6 continued on next page*

*Figure 6 continued*

the TDs of Col1a2$^{creER}$;R26$^{Tom}$;Ptch1$^{fl/fl}$ mice treated with tamoxifen (TAM) at 8 weeks. Dorsal skin was analyzed 9 weeks post TAM treatment (n = 3 mice). No de novo HFs were observed. (**D–G**) Col1a2$^{creER}$;R26$^{Tom}$;Ptch1$^{fl/fl}$ and Col1a2$^{creER}$;R26$^{Tom}$;Ptch1$^{fl/wt}$ control mice were treated with TAM at 8 weeks and dorsal skin was analyzed 10 days, 5 weeks, and 9 weeks after TAM treatment (n = 3 mice per genotype and time point). (**D**) TDs of mice with heterozygous *Ptch1* deletion were phenotypically normal. (**E**) TDs of mice with homozygous *Ptch1* inactivation did not develop de novo HFs. Frequently, a higher cell density in stroma was observed. (**F–G**) Syndecan-1 (SDC1) staining was negative in the condensed stroma of TDs (**F**) as well as in dermal cell condensations (arrows) underneath the IFE and HF infundibula (**G**) in Col1a2$^{creER}$;R26$^{Tom}$;Ptch1$^{fl/fl}$ mice. TD: touch dome. HF: hair follcile. IFE: interfollicular epidermis. DC: dermal cell condensation. Scale bars: 50 µm (**D–G**).

The online version of this article includes the following figure supplement(s) for figure 6:

**Figure supplement 1.** Leakiness in Col1a2$^{creER}$;R26$^{Tom}$;Ptch1$^{fl/fl}$ skin.

**Figure supplement 2.** Stromal Hh-pathway activation in Col1a2$^{creER}$;R26$^{Tom}$;Ptch1$^{fl/fl}$ skin leads to dermal cell condensations.

---

(*Figure 8D*). Reassuringly, in the HF isthmus *Lgr6* expression is restricted to epithelial cells only, whereas *Gli1* is expressed in epithelial and adjacent stromal cells (*Füllgrabe et al., 2015*); suggesting that adjacent epithelial-stromal Hh signaling in areas outside of the TD may form de novo HFs.

## Epithelial and adjacent stromal Hh-pathway activation induces de novo HF formation in hairless paw skin

The mouse hindpaw (plantar) epidermis is a skin region devoid of hair follicles and sweat glands, and is therefore ideal to test whether epithelial and stromal Hh-pathway activation can induce de novo HFs divorced from any confounding effects of nearby HF- or TD-niche signals (*Figure 9A*). When we probed for *Gli1* expression in the plantar skin using *Gli1*$^{LacZ}$ reporter mice, we consistently found small *Gli1*-BGAL expressing clusters of epithelial and adjacent stromal cells in the plantar skin (*Figure 9B*, *Figure 9—figure supplement 1A*), which we confirmed with short-term lineage tracing in Gli1$^{creERT2}$;R26$^{Tom}$ mice (tamoxifen at P8w; sample collection 7 days later) (*Figure 9C*, *Figure 9—figure supplement 1B*). Therefore, the plantar skin was a suitable area for studying if de novo HFs can form in the Gli1 mouse model upon homozygous *Ptch1* inactivation.

Testing for de novo HF induction, we analyzed the hindpaws of Gli1$^{creERT2}$;R26$^{Tom}$;Ptch1$^{fl/fl}$ mice and control littermates (Gli1$^{creERT2}$;R26$^{Tom}$;Ptch1$^{fl/wt}$) using the same treatment scheme and sample collection times as for dorsal skin (tamoxifen at P8w; sample collection 5 and 9 weeks post tamoxifen; additional time points see *Supplementary file 1*). Indeed, we found numerous de novo HFs with hair shafts in the normally hairless region. These HFs were fully Tomato-traced and showed normal inner-layer differentiation based on morphology and K6 staining (*Figure 9D–E*, *Figure 9—figure supplement 2*).

Taken together, the analysis of dorsal HF-isthmus as well as hairless paw skin demonstrated that combined epithelial and stromal Hh-pathway activation can induce de novo HFs independently of TD niches.

## Discussion

Hitherto, hair follicle neogenesis in adult skin had only been observed under exceptional circumstances, such as upon repair of large wounds (*Breedis, 1954*; *Ito et al., 2007*). A recent study found that it is possible to induce HFs even in small wounds upon supra-physiological Hh-pathway activation in the wound stroma (*Lim et al., 2018*). This report and our present study recognize modulation of Hh signaling as a new approach to induce de novo HFs in adult skin, and define activation in the stromal skin compartment as critical. As wounding of skin initiates a major reorganization of the epithelial and mesenchymal tissue including the activation and differentiation of a large number of cell types (*Arwert et al., 2012*; *Joost et al., 2018*; *Schäfer and Werner, 2008*), and may even provide an embryonic-like environment (*Wang et al., 2015*), the precise (e.g. minimal) molecular signals that are required for HF induction in adult skin remain elusive. Here we revealed that in unwounded skin, experimentally elevated Hh signaling in epithelial and adjacent stromal cells was sufficient to induce

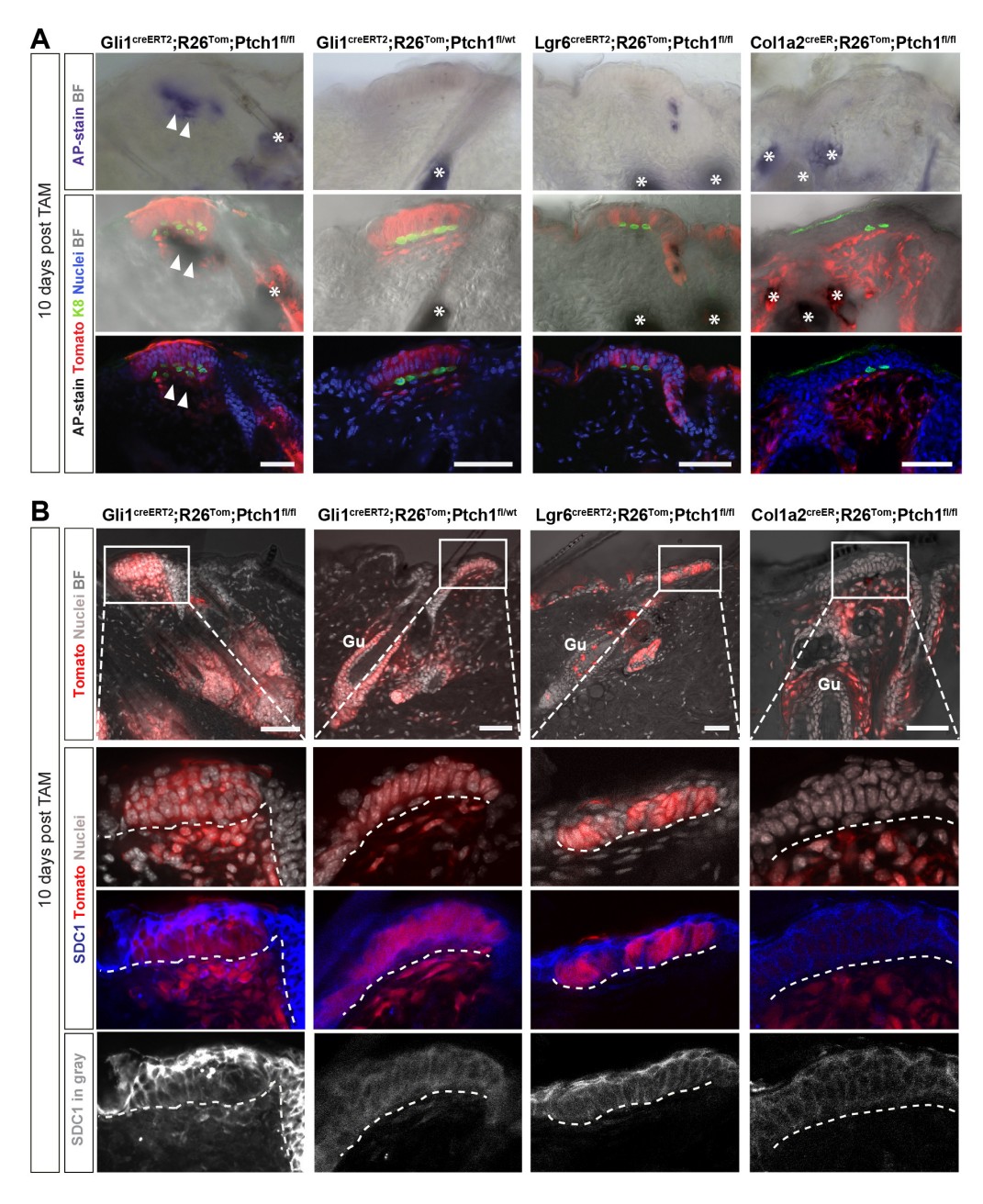

**Figure 7.** Expression of markers indicative for hair follicle (HF)-induction competent stroma in Gli1[creERT2];R26[Tom];Ptch1[fl/fl] touch domes (TDs). (A–B) Mice were treated with tamoxifen (TAM) at 8 weeks and dorsal skin was analyzed 10 days later using the alkaline phosphatase (AP) enzymatic assay (A; n = 2–3 mice per genotype) or Syndecan-1 (SDC1) immunofluorescence staining (B; n = 3 mice per genotype). (A) The TD stroma of Gli1[creERT2];R26[Tom];Ptch1[fl/fl] skin stained AP-positive, indicating pre-dermal condensate formation. Please note that the nuclear staining (DAPI) in the areas of AP-positive signal is present but very dim (quenched). The TD stroma in skin of Gli1[creERT2];R26[Tom];Ptch1[fl/wt], Lgr6[creERT2];R26[Tom];Ptch1[fl/fl] and Col1a2[creER];R26[Tom];Ptch1[fl/fl] stained AP-negative. (B) TD stroma of Gli1[creERT2];R26[Tom];Ptch1[fl/fl] skin was clearly positive for SDC1 staining, while the TD stroma of Gli1[creERT2];R26[Tom];Ptch1[fl/wt] skin showed very weak to negative SDC1 staining. In the TDs of Lgr6[creERT2];R26[Tom];Ptch1[fl/fl] and Col1a2[creER];R26[Tom];Ptch1[fl/fl] skin, SDC1 staining was absent. Gu: Guard hair. TAM: tamoxifen. BF: bright field. Asterisks: sebaceous glands stain positive for AP. Arrowheads: positive AP-staining in TD stroma. Dashed line: epithelial-stromal border. Scale bars: 50 µm (A–B).

The online version of this article includes the following figure supplement(s) for figure 7:

**Figure supplement 1.** Capturing very early epithelial bud formation in TDs using BrdU incorporation.

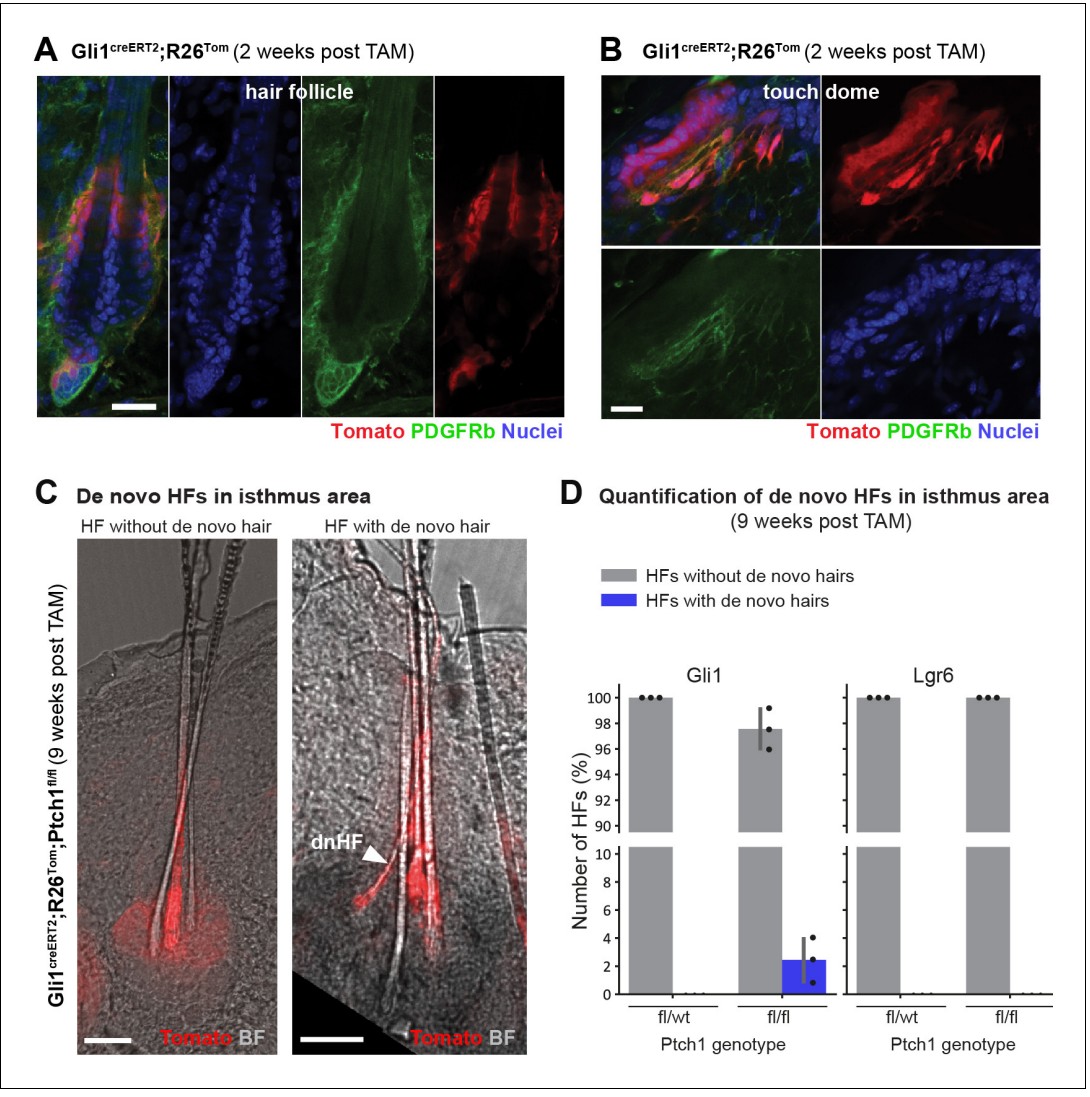

**Figure 8.** De novo hair follicle (HF) formation in isthmus area of pre-existing HFs. (**A–B**) PDGFRb (CD140b antibody) staining of Gli1$^{creERT2}$;R26$^{Tom}$ mice, treated with tamoxifen (TAM) at 8 weeks of age and traced for 2 weeks. Tomato-traced PDGFRb+ stromal cells are present in the HF isthmus area (**A**) and in the touch dome (TD) (**B**). (**C–D**) Mice were treated with TAM at 8 weeks of age and dorsal skin was analyzed 9 weeks post TAM. (**C**) Gli1$^{creERT2}$;R26$^{Tom}$;Ptch1$^{fl/fl}$ dorsal skin. Left panel: HFs without de novo HF contain three hair shafts (3 rounds of anagen). Right panel: HFs with de novo HF often contain four hair shafts including one thinner hair shaft with bent shape (arrowhead). (**D**) Quantification of de novo HFs in the isthmus area of Gli1$^{creERT2}$;R26$^{Tom}$;Ptch1$^{fl/fl}$ (n = 668 HFs from n = 3 mice), Lgr6$^{creERT2}$;R26$^{Tom}$;Ptch1$^{fl/fl}$ (n = 207 HFs from n = 3 mice), Gli1$^{creERT2}$;R26$^{Tom}$;Ptch1$^{fl/wt}$ (n = 335 HFs from n = 3 mice) and Lgr6$^{creERT2}$;R26$^{Tom}$;Ptch1$^{fl/wt}$ (n = 83 HFs from n = 3 mice) (***Figure 8—source data 1***). De novo HFs were only detected in isthmus areas of Gli1$^{creERT2}$;R26$^{Tom}$;Ptch1$^{fl/fl}$ mice. p-value=0.09 (t-test comparing Gli1$^{creERT2}$;R26$^{Tom}$;Ptch1$^{fl/fl}$ and Gli1$^{creERT2}$;R26$^{Tom}$;Ptch1$^{fl/wt}$). n = 3 mice (**A–C**). dnHF: de novo hair follicle. TAM: tamoxifen. BF: bright field. Scale bars: 20 μm (**A**), 10 μm (**B**), 50 μm (**C**).

The online version of this article includes the following source data for figure 8:

**Source data 1.** Quantification of de novo hairs from isthmus.

de novo HFs, extending our understanding of how Hh signaling can be modulated to induce HFs in adult skin.

To achieve efficient de novo HF induction in unwounded skin, supra-physiological Hh signaling in both compartments, the epithelium and stroma, was necessary. Normal TD maintenance also requires active and balanced Hh signaling in adjacent epithelial and stromal cells (***Figure 4E***; ***Xiao et al., 2015***). Increased Hh-signaling levels in TD epithelial cells result in BCC-like tumors, even

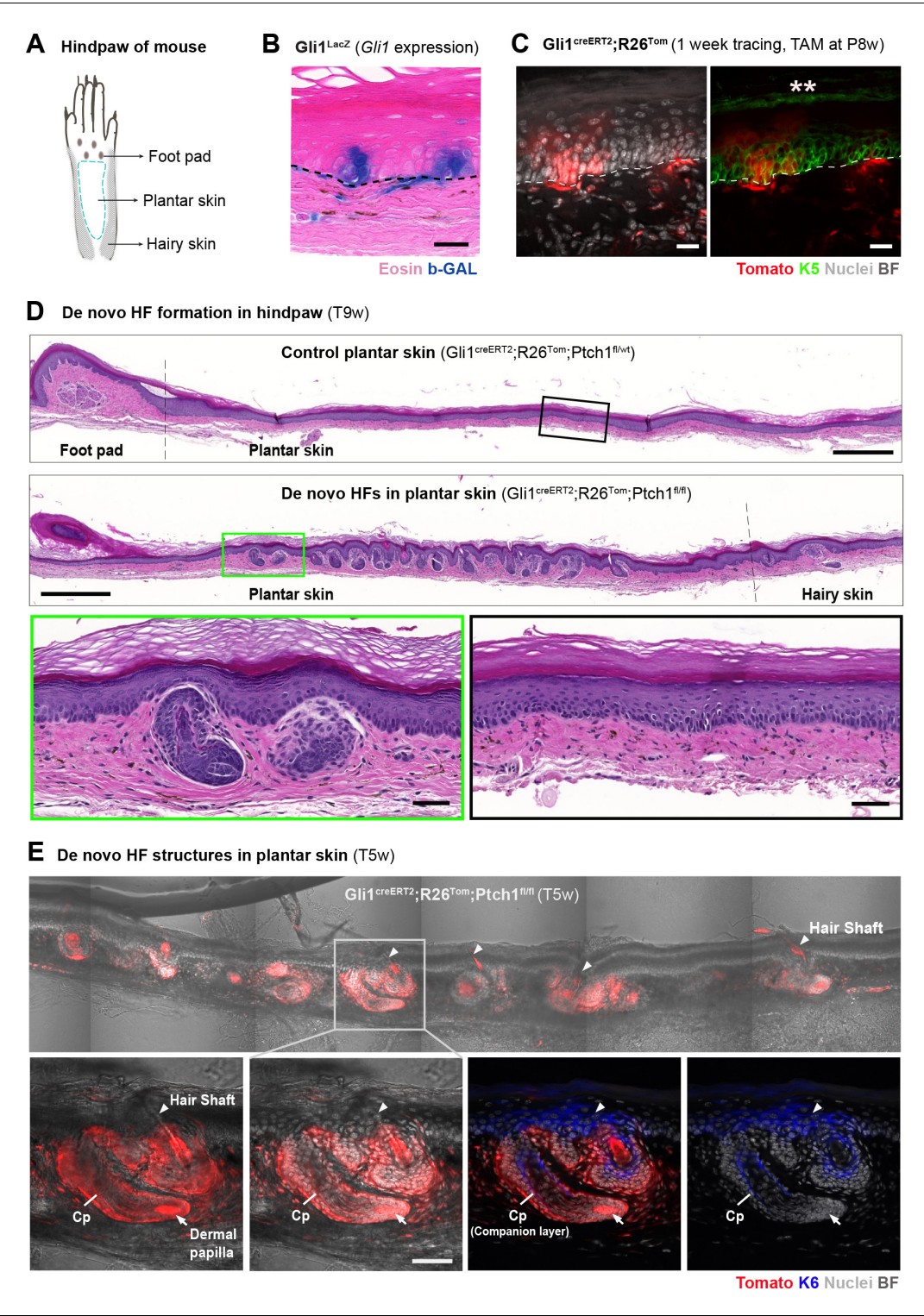

**Figure 9.** Formation of de novo hair follicles (HFs) in the plantar skin of Gli1^creERT2;R26^Tom;Ptch1^fl/fl mice. (**A**) Illustrative cartoon of a mouse hindpaw. (**B**) *Gli1*^LacZ expression in the plantar skin (n = 3 mice). (**C**) Gli1^creERT2; R26^Tom mice were treated with tamoxifen (TAM) at 8 weeks of age. The hindpaws were collected 1 week post TAM and immuno-stained with K5 antibody (n = 3 mice). Asterisks mark autofluorescence on the outermost keratinized layer. (**D–E**) Gli1^creERT2;R26^Tom;Ptch1^fl/fl and control Gli1^creERT2;R26^Tom;Ptch1^fl/wt mice were treated with TAM at 8 weeks of age. Hindpaws were collected 5 or 9 weeks post TAM (n = 3 mice for each genotype; except n = 2 for Gli1^creERT2;R26^Tom;Ptch1^fl/wt 5 weeks post TAM ). (**D**) Numerous de novo HFs formed in

*Figure 9 continued on next page*

*Figure 9 continued*

the plantar skin of Gli1$^{creERT2}$;R26$^{Tom}$;Ptch1$^{fl/fl}$ mice, while the same skin region in the control mice remained phenotypically normal. Green and black frames: zoom-in of plantar epidermis. Hematoxylin and eosin stained. (E) De novo HF structure in plantar skin of Gli1$^{creERT2}$;R26$^{Tom}$;Ptch1$^{fl/fl}$ mice are fully Tomato-traced and contain a K6+ companion layer as well as hair shafts (n = 2 mice). Arrowheads: de novo hair shafts. Arrows: dermal papilla. Cp: companion layer. Dashed line: epithelial-stromal border. TAM: tamoxifen. BF: bright field. Scale bars: 20 μm (**B–C**), 500 μm (**D** panaromas), 50 μm (**D, E** insets).

The online version of this article includes the following figure supplement(s) for figure 9:

**Figure supplement 1.** Physiological Hh/Gli signaling in the hindpaw plantar skin.
**Figure supplement 2.** De novo HF formation in hindpaw of Gli1$^{creERT2}$;R26$^{Tom}$;Ptch1$^{fl/fl}$ mice.

though stromal cells have active (albeit physiological) Hh signaling. Only in extremely rare cases (twice) did we detect a de novo HF-like structure in TDs of Lgr6$^{creERT2}$;R26$^{Tom}$;Ptch1$^{fl/fl}$ mice based on morphology (as lineage tracing in these mice cannot provide information on de novo HF formation; *Figure 1H* and *Figure 1—figure supplement 2*). Importantly therefore, to effectively induce de novo HFs in TDs, high levels of Hh/Gli signaling in both compartments were necessary (Gli1$^{creERT2}$; R26$^{Tom}$;Ptch1$^{fl/fl}$). This requirement of Hh-signal activation at precise levels and in the right compartments is in agreement with a recent study demonstrating that β-catenin-induced de novo HF formation was not only dependent on stromal Hh signaling, but also required two intact *Smo* alleles (for a maximal Hh-pathway activation) to enable efficient de novo HF induction (*Lichtenberger et al., 2016*).

It has been shown more than twenty years ago that the activation of epithelial β-catenin in mouse skin can induce new HFs (*Gat et al., 1998*; *Lo Celso et al., 2004*), and more recently that activation of epithelial Wnt/β-catenin signaling increases de novo HF formation within wounds (*Ito et al., 2007*). However, Wnt signaling has to be blocked in dermal fibroblasts to allow de novo HF induction during wound regeneration (*Rognoni et al., 2016*). It is known that early stage BCCs resemble early stages of HF development and both are dependent on Wnt- and Hh-pathway activation, with the major morphological difference that BCC lacks a dermal condensate (*Yang et al., 2008*). Learning from abrogated embryonic HF development (*St-Jacques et al., 1998*) led us to hypothesize that simultaneously activating supra-physiological Hh signaling in the stroma underneath developing BCC may enable de novo HFs. Indeed, by coordinating the activation of Hh/Gli signaling (cell type specific and high levels) we were able to induce de novo HFs by 'redirecting' some of the BCC buds to HF formation without the need of wounding. Nevertheless, this induction occurred in the presence of oncogenic signal activation (i.e. presence of a tumor environment or tumor-like cellular status of Ptch1$^{fl/fl}$ HF-inducing epithelial cells) which may to some extent mimic a wounding situation (*Dvorak, 1986*).

The molecular and cellular similarities of tumorigenesis and wound healing are still unfolding, yet whenever de novo HFs were to be found, either oncogenic signaling or a wound environment was involved. This supports the long-standing recognition of the similarity between tumor and wound healing signaling (tumors as 'wounds that do not heal') – and raises the key question of what exactly is the relationship between tumorigenesis and signals inducing de novo HF formation? It may indeed be the case that in order to overcome inhibitory signals, de novo HF morphogenesis in adult skin requires such major activating signals provided by oncogenesis or wounding. Interestingly however, previous literature suggests that only initial HF placode and/or dermal condensate formation may require such strong signals whereas progression to a mature HF does not require continued tumorigenic or wound signaling (*Brown et al., 2017*; *Ito et al., 2007*; *Lo Celso et al., 2004*; *Silva-Vargas et al., 2005*). For example, HF tumors require continuous Wnt/β-catenin signaling, whereas transient activation of this pathway is sufficient to induce de novo HFs in adult mouse epidermis (*Lo Celso et al., 2004*). More recently, intra vital imaging from Wnt/β-catenin induced tumor outgrowths demonstrated that non-mutant cells, remaining from regressed outgrowths, could develop into new functional HFs. Most interestingly, tumor outgrowth depended on the presence of mutated cells, however the new appendages were formed from wild type cells facilitated by their (altered) niche environment (*Brown et al., 2017*). Taken together, these are promising examples that de novo HF induction without accompanied tumor growth in unwounded adult skin may in principle be

possible, if the right signals at the right time and restricted period, and in the right compartments were provided. Here, we spatiotemporally defined such productive and specific molecular signals.

Lastly, and importantly, we exploited the hairless paw plantar skin to examine de novo HF morphogenesis in the absence of confounding signals from pre-existing HFs or TDs. Strikingly, we observed numerous de novo HFs throughout this nominally hairless skin in Gli1$^{creERT2}$;R26$^{Tom}$;Ptch1$^{fl/fl}$ mice. Crucially, nearly all these de novo HFs developed without attendant BCC-like lesions suggesting that de novo HF morphogenesis may indeed be successfully initiated without a tumor microenvironment; while the vismodegib experiment suggests persistence of such structures when the tumor microenvironment regresses in dorsal skin. Examining these two divergent tissues in molecular detail, one permissive (dorsal) and one suppressive (paw) to dual BCC and HF induction, could therefore be a next step of unraveling the complexity of how these heterogeneous signals interact.

In sum, molecular strategies for the induction of complex epithelial structures in the adult remain a major challenge in regenerative medicine. Our study demonstrates that cell-type specific modulation of a single pathway was sufficient to induce complex epithelial structures in the adult body, a discovery aiding our understanding of adult tissue biology and regenerative medicine.

# Materials and methods

## Key resources table

| Reagent type (species) or resource | Designation | Source or reference | Identifiers | Additional information |
|---|---|---|---|---|
| Genetic reagent (*M. musculus*) | Lgr6-EGFP-IRES-CreERT2 | *Snippert et al., 2010* | Jackson Labs stock no: 016934 RRID:IMSR_JAX:016934 | Received from H Clevers |
| Genetic reagent (*M. musculus*) | Gli1-LacZ | *Bai et al., 2002* | Jackson Labs stock no: 008211 RRID:IMSR_JAX:008211 | Received from F Aberger |
| Genetic reagent (*M. musculus*) | Gli1-CreERT2 | *Ahn and Joyner, 2004* | Jackson Labs stock no: 007913 RRID:IMSR_JAX:007913 | Received from F Aberger |
| Genetic reagent (*M. musculus*) | Col1a2-CreER | *Zheng et al., 2002* | Jackson Labs stock no: 029567 RRID:IMSR_JAX:029567 | Received from B Crombrugghe |
| Genetic reagent (*M. musculus*) | R26-tdTomato | *Madisen et al., 2010* | Jackson Labs stock no: 007908 RRID:IMSR_JAX:007908 | Obtained from Jackson Laboratory |
| Genetic reagent (*M. musculus*) | Ptch1neo(fl)Ex2(fl) | *Kasper et al., 2011* | | Received from S. Teglund |
| Genetic reagent (*M. musculus*) | C57BL/6J | | Jackson Labs stock no: 000664 RRID:IMSR_JAX:000664 | Received from Preclinical Laboratory Karolinska Institutet |
| Antibody | Rabbit polyclonal anti-K5 | BioLegend | Cat# PRB-160P RRID:AB_291581 | (1:1000) |
| Antibody | Guinea pig polyclonal anti-K5 | US Biological | Cat# C9097-37 RRID:AB_2134285 | (1:50) (1:200 for RNA-FISH) |
| Antibody | Rat monoclonal anti-SDC1 | BD Biosciences | Cat# 553712 RRID: AB_394998 | (1:500) |
| Antibody | Rat monoclonal anti-K8 | DSHB (Developmental Studies Hybridoma Bank) | Cat# TROMA-I RRID:AB_531826 | (1:1000) |
| Antibody | Rabbit polyclonal anti-K6 | BioLegend | Cat# PRB-169P RRID:AB_10063923 | (1:2000) |
| Antibody | Guinea pig polyclonal anti-K71 | Progen | Cat# GP-K6irs1 RRID:AB_2716781 | (1:100) |
| Antibody | Rabbit polyclonal anti-RFP | Rockland | Cat# 600-401-379 RRID:AB_2209751 | (1:100) |

*Continued on next page*

*Continued*

| Reagent type (species) or resource | Designation | Source or reference | Identifiers | Additional information |
|---|---|---|---|---|
| Antibody | Rat monoclonal anti-BrdU | Serotec | Cat# OBT0030G RRID: AB_609567 | (1:400) |
| Antibody | Rabbit polyclonal anti-GFP | Thermo Fisher Scientific | Cat# A-11122 RRID:AB_221569 | (1:500) |
| Antibody | Rabbit polyclonal anti-Ki67 | Abcam | Cat# ab15580 RRID:AB_443209 | (1:200) |
| Antibody | Rat monoclonal anti-CD140b | BioLegend | Cat# 136008 RRID:AB_2268091 | (1:100) |
| Chemical compound, drug | TO-PRO-3 | Invitrogen | Cat# T3605 | (1:1000) |
| Chemical compound, drug | Hoechst 33342 | Invitrogen | Cat# H3570 | (1 µg/mL) |
| Chemical compound, drug | DAPI | Invitrogen | Cat# D1306 | (1 µg/mL) |
| Chemical compound, drug | Tamoxifen | Sigma | Cat# T5648 | |
| Chemical compound, drug | 4-OH Tamoxifen | Sigma | Cat# H6278 | |
| Chemical compound, drug | NBT/BCIP stock solution | Roche | Cat# 11681451001 | |
| Chemical compound, drug | Vismodegib, Free Base | LC Laboratories | Cat# V-4050 | |
| Commercial assay or kit | RNAscope Multiplex Fluorescent Kit v2 | ACDBio/Bio-Techne | Cat# 323100 | |
| Commercial assay or kit | TSA Cy 3, Cy 5, TMR, Fluorescein Evaluation Kit | Perkin Elmer | Cat# NEL760001KT | |
| Sequence-based reagent | 3-plex Positive Control Probe | ACDBio/Bio-Techne | Cat# 320881 | RNA-FISH Probe |
| Sequence-based reagent | 3-plex Negative Control Probe | ACDBio/Bio-Techne | Cat# 320871 | RNA-FISH Probe |
| Sequence-based reagent | Mm-Gli1 | ACDBio/Bio-Techne | Cat# 311001 | RNA-FISH Probe |

## Mouse models and treatments

Gli1$^{creERT2}$;R26$^{Tom}$;Ptch1$^{fl/fl}$ mice and control littermates, and Lgr6$^{creERT2}$;R26$^{Tom}$;Ptch1$^{fl/fl}$ mice and control littermates were treated with tamoxifen at second telogen (mice aged 8 weeks, 6 mg tamoxifen i.p. in corn oil, 20 mg/mL). Dorsal samples were taken at different time points after tamoxifen treatment (as indicated in the text), and were obtained via 3–4 mm full thickness biopsies or by sacrificing the animal (n $\geq$ 3 mice for each genotype; for details please see *Supplementary file 1*). Some of the Col1a2$^{creER}$;R26$^{Tom}$;Ptch1$^{fl/fl}$ mice, when receiving the same treatment as the Gli1 and Lgr6 models as described above, developed within a few weeks a severe intestinal phenotype, precluding comparative skin analysis at 5 and 9 weeks post tamoxifen administration. Thus Col1a2$^{creER}$;R26$^{Tom}$;Ptch1$^{fl/fl}$ mice and control littermates were treated at second telogen topically or with a reduced amount of tamoxifen i.p. (mice aged 8 weeks, 2 × 0.75 mg 4-OH tamoxifen topically or 3 mg tamoxifen i.p. in corn oil, 20 mg/mL; *Supplementary file 1*), and some were treated at first telogen (mice aged 3 weeks, 1 mg tamoxifen i.p. in corn oil, 20 mg/mL). Samples were harvested at different time points after tamoxifen treatment (as indicated in the text). More details of analyzed mice, including control mouse experiments, treatments, and phenotypes, are given in *Supplementary file 1*. For BrdU incorporation, mice were injected i.p. with BrdU (Sigma) solution (10 mg/ml). A dose of 0.1 mg/g body weight was administered 2 hr prior to sacrifice. For vismodegib treatment, the Gli1$^{creERT2}$;R26$^{Tom}$;Ptch1$^{fl/fl}$ mice and control littermates Gli1$^{creERT2}$;R26$^{Tom}$;Ptch1$^{fl/wt}$ were treated with tamoxifen at 8 weeks of age. Five to seven weeks after tamoxifen

treatment, when de novo HFs were clearly established in TDs, a dorsal biopsy prior to vismodegib treatment was taken. From then, the vismodegib was given daily (50 mg/kg body weight i.p.) for a week and dorsal samples were collected for further analysis (n ≥ 3 mice). Embryos were collected at E15.5 and E16.5, and wild-type control samples for dorsal tissue were collected at postnatal day 27 and week 9. All mouse experiments were performed in accordance to Swedish legislation and approved by the Stockholm South or Linköping Animal Ethics Committees.

## Tissue staining, microscopy and image analysis

All antibodies, β-Galactosidase, alkaline phosphatase (AP), and RNA-FISH stainings were performed on either mouse dorsal skin or hindpaw samples as described below (1-5). For nuclear stains, TO-PRO-3, Hoechst 33342 or DAPI (all from Invitrogen) were used in the different applications below (1-4).

(1) Immunofluorescence (IF) staining on formalin-fixed, paraffin-embedded (FFPE) sections. After de-waxing and antigen retrieval in 10 mM citrate buffer or Diva Decloaker (Biocare Medical) for approximately 20 min in a pressure cooker (2100 Retriever), the sections were blocked with serum and then incubated with primary antibodies specific for K5 (rabbit 1:1000), Syndecan-1 (1:500). Secondary antibodies were Alexa Fluor Dyes (Invitrogen, 1:500). Used in *Figure 4A*.

(2) Horizontal whole mount (HWM) IF staining. Samples of dorsal skin were fixed in 4% PFA for 20 min and mounted in OCT embedding medium (Histolab). Subsequently, 60–150 µm sections were cut with a cryostat, blocked with PB buffer (0.1% fish skin gelatin, 0.5% Triton X-100% and 0.5% skimmed milk powder in PBS) and stained as described previously (*Driskell et al., 2009*). The nuclear stains were applied at the same time as secondary antibodies. Primary antibodies used: K8 (1:1000), K5 (rabbit 1:1000, guinea pig 1:50), K6 (1:2000), K71 (1:100), Syndecan-1 (1:500), GFP (1:500), BrdU (1:400), RFP (1:100), Ki67 (1:200), CD140b (1:100). Secondary antibodies used: Alexa Fluor Dyes 488, 546, 647 or 680 (Invitrogen, 1:500). Used in: *Figure 1D,F–I*; *Figure 1—figure supplements 1*, *2* and *3*; *Figure 2A,C*; *Figure 2—figure supplements 1* and *2*; *Figure 3A–F*; *Figure 4B,C*; *Figure 5C,D*; *Figure 5—figure supplement 1*; *Figure 6D–G*; *Figure 6—figure supplements 1* and *2*; *Figure 7A, B*; *Figure 7—figure supplement 1*; *Figure 8A,B*; *Figure 9C,E*; *Figure 9—figure supplements 1B* and *2A–B*.

(3) RNA Fluorescent in situ hybridization (RNA-FISH). RNA-FISH for *Gli1* was performed using the RNAscope Multiplex Fluorescent Detection Kit v2 according to manufacturer's instructions using TSA with Fluorescein on 4-10 µm FFPE sections. All sections were co-stained with anti-RFP (1:100) and anti-K5 (guinea pig 1:200) antibodies and counterstained with DAPI (1 µg/mL). Each co-staining was performed on two different mice for each time point, with the exception of only one mouse for the T10d time point. Used in: *Figure 4D–G*; *Figure 4—figure supplement 1*.

(4) Alkaline phosphatase (AP) enzymatic assay. HWM skin samples were cut into 60–100 µm sections, fixed in acetone for 1 hr at 4˚C, pre-incubated with NTMT solution (100 mM NaCl, 100 mM Tris-Cl pH9.5, 50 mM MgCl2, 0.1% Tween-20) at RT for 10 min and stained in NBT/BCIP solution (20 µL in 1 ml NTMT solution) for 1–3 min at 37˚C. The reaction was stopped with 20 mM EDTA in PBS. The AP-stained tissue subsequently underwent HWM IF staining. Used in: *Figure 7A*.

(5) LacZ (β-Galactosidase) staining. Freshly obtained skin tissue was fixed (4% paraformaldehyde in PBS) for 30 min at RT. Tissues were washed for 15 min with rinse buffer (2 mM MgCl2, 0.01% Nonidet P-40 in PBS). Subsequently, the β-galactosidase substrate solution (1 mg/mL X-Gal, 5 mM $K_3Fe$ $(CN)_6$, 5 mM $K_4Fe$ $(CN)_6·3H_2O$ in rinse buffer) was added and the tissues were incubated for 18 hr at 37˚C in the dark. The substrate was removed, and the tissues were washed twice in PBS for 10 min and kept in 70% ethanol until embedding (maximum 48 hr). The stained tissues were processed into paraffin blocks according to standard procedures. Tissue sections (4 µm) were prepared and counterstained with eosin or H&E. Used in *Figure 1E*; *Figure 9B*; *Figure 9—figure supplement 1A*.

Imaging was performed using a Leica (color bright-field images), LSM710-NLO confocal microscope (Zeiss) or Nikon A1R confocal microscope. Image analysis was performed using NIS-Elements software (Nikon), Zen 2009 software (Zeiss), or ImageJ, and images were occasionally optimized for brightness, contrast, and color balance. RNA-FISH images are presented as maximum intensity projections covering 6 µm of depth.

## Measurements and quantitative image analysis of de novo hair follicles

Measurements of de novo and pre-existing Zig-zag HFs were performed in samples from Gli1$^{creERT2}$;R26$^{Tom}$;Ptch1$^{fl/fl}$ and Gli1$^{creERT2}$;R26$^{Tom}$;Ptch1$^{fl/wt}$ mice taken 9 weeks after tamoxifen treatment. Image J was used to analyze the images.

(1) Quantification of de novo HFs in touch domes. Touch domes were identified via Tomato-tracing and the presence of Merkel cells (K8+). Subsequently, all HFs and hair shafts were visualized with bright field images. Only telogen stage hairs that were fully visible from the bulge to the IFE opening were selected and used to measure their length (i.e. the distance from the bottom/hair club of the telogen hair shaft to the opening of the follicle in the IFE), and width (which was measured where the hair shaft meets the IFE); see *Figure 2C*. In total, de novo HFs were observed in all of the 27 analyzed touch domes (on average 2.6 de novo HFs per touch dome) in three different mice, and 34 hair shafts from 20 touch domes qualified for measurements. As controls, pre-existing Zig-zag hairs of Gli1$^{creERT2}$;R26$^{Tom}$;Ptch1$^{fl/fl}$ (i.e. the same samples as for de novo HF measurements; n = 3 mice; 314 hairs in total) and Zig-zag hairs of Gli1$^{creERT2}$;R26$^{Tom}$;Ptch1$^{fl/wt}$ control mice lacking BCC and de novo HFs (n = 3 mice; 437 hairs in total) were analyzed in the same way (*Figure 2—source data 1*).

To statistically estimate whether the putative de novo HFs could merely represent small Zig-zag hairs, a bootstrapping approach was used: a random sample corresponding to the number of observed de novo HFs (n = 34) was taken from the distribution of observed Zig-zag hairs (n = 751) for 10.000.000 times, and the probability that this random sample is equal to or smaller in length and width than the de novo HFs was subsequently calculated.

(2) Quantification of de novo hairs in isthmus areas of pre-existing HFs. Quantification of de novo hairs in the isthmus area of HFs (note that we were not able to tell Zig-zag/Awl/Au HFs apart and collectively call them Zig-zag) was performed in samples from Gli1$^{creERT2}$;R26$^{Tom}$;Ptch1$^{fl/fl}$, Gli1-$^{creERT2}$;R26$^{Tom}$;Ptch1$^{fl/wt}$, Lgr6$^{creERT2}$;R26$^{Tom}$;Ptch1$^{fl/fl}$ and Lgr6$^{creERT2}$;R26$^{Tom}$;Ptch1$^{fl/wt}$ mice (n = 3 for each genotype) taken 9 weeks after tamoxifen treatment. De novo hairs were identified by their relatively smaller size, growing pattern and/or there being more than three hairs in a pre-existing HF. De novo hairs growing from the isthmus area were only observed in the Gli1$^{creERT2}$;R26$^{Tom}$;Ptch1$^{fl/fl}$ mice with a total of 16 identified de novo hairs in 668 Zig-zag HFs. Gli1$^{creERT2}$;R26$^{Tom}$;Ptch1$^{fl/wt}$ (335 HFs), Lgr6$^{creERT2}$;R26$^{Tom}$;Ptch1$^{fl/fl}$ (207 HFs) and Lgr6$^{creERT2}$;R26$^{Tom}$;Ptch1$^{fl/wt}$ (83 HFs) mice showed no de novo hairs from the isthmus area.

## Acknowledgements

We thank Beate Lichtenberger and Rune Toftgård for valuable discussions and feedback on the manuscript, Sunny Wong and Scott Atwood for sharing important scientific insights, Maryam Saghafian for her help with mouse work, Mayumi Ito and Chae Ho Lim for their advice to perform the alkaline phosphatase assay, Maria Hölzl for embryonic mouse skin sections, Alexandra Joyner for sharing the Gli1$^{creERT2}$ and Gli1$^{LacZ}$ mice, Hans Clevers for sharing the Lgr6$^{creERT2}$ mice, and Benoit de Crombrugghe for sharing the Col1a2$^{creER}$ mice. This work was supported by grants from the Swedish Cancer Society, Swedish Research Council, Swedish Foundation for Strategic Research, Center for Innovative Medicine, and Ragnar Söderberg Foundation to MK, the Swedish Cancer Society, and Swedish Society for Medical Research to MG, the Wenner-Gren Foundation to TD, and Karolinska Institutet KID funding to SJ, TJ and KA. Parts of this study were performed at the Live Cell Imaging facility/Nikon Center of Excellence, Department of Biosciences and Nutrition, Karolinska Institutet. The authors have no conflicts of interest to declare.

## Additional information

### Funding

| Funder | Grant reference number | Author |
| --- | --- | --- |
| Cancerfonden | CAN 2011/1180 | Maria Kasper |
| Ragnar Söderbergs stiftelse | M127/12 | Maria Kasper |
| Stiftelsen för Strategisk Forskning | FFL12-0133 | Maria Kasper |

| Karolinska Institutet | KID funding | Karl Annusver Tina Jacob Simon Joost |
| --- | --- | --- |
| Cancerfonden | CAN 2014/1376 | Marco Gerling |
| Vetenskapsrådet | 2018-02963 | Maria Kasper |
| Wenner-Gren Foundation | UPD2017-0264 | Tim Dalessandri |
| Swedish Society for Medical Research | | Marco Gerling |
| Cancerfonden | CAN 2018/793 | Maria Kasper |

The funders had no role in study design, data collection and interpretation, or the decision to submit the work for publication.

## Author contributions
Xiaoyan Sun, Conceptualization, Data curation, Validation, Investigation, Visualization, Methodology, Writing - original draft, Writing - review and editing; Alexandra Are, Conceptualization, Data curation, Validation, Investigation, Visualization, Methodology, Writing - review and editing; Karl Annusver, Data curation, Formal analysis, Validation, Investigation, Visualization, Methodology, Writing - review and editing; Unnikrishnan Sivan, Data curation, Validation, Investigation, Visualization, Methodology; Tina Jacob, Validation, Investigation, Visualization, Writing - review and editing; Tim Dalessandri, Validation, Investigation, Writing - review and editing; Simon Joost, Formal analysis, Validation, Investigation; Anja Füllgrabe, Data curation, Investigation, Methodology; Marco Gerling, Resources, Data curation, Writing - review and editing; Maria Kasper, Conceptualization, Data curation, Supervision, Funding acquisition, Validation, Investigation, Visualization, Methodology, Writing - original draft, Project administration, Writing - review and editing

## Author ORCIDs
Xiaoyan Sun (iD) https://orcid.org/0000-0002-1179-9570
Karl Annusver (iD) https://orcid.org/0000-0002-9515-7216
Unnikrishnan Sivan (iD) https://orcid.org/0000-0002-6074-6299
Tina Jacob (iD) https://orcid.org/0000-0002-7180-2698
Tim Dalessandri (iD) https://orcid.org/0000-0001-5690-295X
Anja Füllgrabe (iD) http://orcid.org/0000-0002-8674-0039
Maria Kasper (iD) https://orcid.org/0000-0002-6117-2717

## Ethics
Animal experimentation: All animal work was approved and permitted by the Local Ethical Committee on Animal Experiments (Stockholm South Animal Ethics Committee; permit number S40-13, Linköping Animal Ethics Committee; permit number 79-15) and conducted according to The Swedish Animal Agency's Provisions and Guidelines for Animal Experimentation recommendations.

## Decision letter and Author response
Decision letter https://doi.org/10.7554/eLife.46756.sa1
Author response https://doi.org/10.7554/eLife.46756.sa2

# Additional files

## Supplementary files
• Supplementary file 1. List of mouse experiments. Given are the following details per mouse: mouse genotype, treatment, analysis time point, tissue sampling (biopsy or final material), comments/observations, mouse ID.

• Transparent reporting form

## Data availability

All data generated or analyzed during this study are included in the manuscript and supporting files. Source data files have been provided for Figures 2 and 8.

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
