## [Decision Letter]

**Acceptance summary:**

The revised manuscript by Sun and Are et al. contains a tour de force of new experiments aimed at showing where de novo hair follicle formation can occur in mouse skin and exploring whether these follicles are similar to existing follicles. The data and conclusions are quite convincing, highly relevant and impactful in the current skin field.

**Decision letter after peer review:**

Thank you for submitting your article "Coordinated hedgehog signaling induces new hair follicles in adult skin" for consideration by *eLife*. Your article has been reviewed by two peer reviewers, and the evaluation has been overseen by a Reviewing Editor and Marianne Bronner as the Senior Editor. The reviewers have opted to remain anonymous.

The reviewers have discussed the reviews with one another and the Reviewing Editor has drafted this decision to help you prepare a revised submission.

Summary:

Sun et al. use inducible reporter mice to show that localized Hedgehog signaling can induce de novo hair follicle formation in adult mice. They show that diffuse Hedgehog activation via loss of Ptch1 in the epidermis (Lgr6-CreERT2) or the dermis (Col1a2-CreER) does not induce de novo hair follicle formation, but a localized source of Hedgehog activation via loss of Ptch1 at the touch dome (Gli1-CreERT2) does. The hair follicles show at least some of the necessary layers of a fully formed structure such as the inner root sheath (Krt71) and outer root sheath (Krt6), and seem to be fully functional as they produce hair. The de novo follicles are also associated with dermal condensates (Sdc1 and Alpl), a structure necessary for placode formation and hair follicle growth. This work complements the work by Lim et al., 2018, who show that Hedgehog can stimulate de novo hair follicles during wound healing. This work is novel and would be of broad interest to the readers of *eLife* both in the instructive signals that organs need to develop (or regenerate) and as a potential way to induce de novo hair follicles in adult humans (if one gets past the concomitant induction of basal cell carcinoma).

Essential revisions:

1) It would be interesting to discuss whether tumorigenic microenvironment or cellular status mimics wounding situations.

2) Some parts in the manuscript including the Abstract may be misleading to provide an impression that the combination of epithelial and dermal Hh activation leads to HF, while epithelial Hh alone leads to tumor. The data shows that the Hh activation in both epidermal and dermal cells also results in BCC, but with hair follicle induction. Hair follicles are immature and abnormal and appear to be a part of the tumor. The authors should revise the text/Abstract to clearly indicate this important point.

3) Since Lgr6+ epithelial cells are known as stem cells, do they form hair follicles upon Hh activation in unwounded skin regardless of TD environment? In Figure 1E, Lgr6+ epithelial cells exist beyond TD. What if Hh is activated in the Lgr6+cells outside of TD and underlying stroma using Lgr6^CreER^/Col1a2^CreER^/Ptch^fl/fl^?

4) Do dermal condensates form only in the stroma of TD? If this is the case, the authors may be able to show that the TD area is specialized to form hair follicles.

5) In Figure 1D, Gli1 is positive in follicular epithelium and surrounding stromal cells of guard hairs. When Hh is activated in Gli1/Tom/Ptch1^fl/fl^, did de novo hair follicles form in the Gli1+ area of guard hairs? It may help support that TD is a primed structure to form hair follicles in adult unwounded mouse skin.

6) Although this work clearly shows that localized Hedgehog signaling can induce hair follicles, the equivalency of these structures to existing follicles and the lack of mechanism should be addressed as detailed below.

7) The de novo hair follicles form under constant Hedgehog signaling, which is typically only active during the anagen phase of the hair follicle and is shut down to allow the hair follicle to proceed to catagen/telogen.

Do the de novo hair follicles cycle when induced with the Gli1-CreERT2 promoter?

8) Is Hedgehog pathway activation necessary for de novo hair follicle formation or merely Ptch1 loss?

9) Can the hair follicles persist without Hedgehog signaling?

Use of Smo inhibitors such as vismodegib can be helpful to answer these questions by administering the drug after tamoxifen injection or after the hair follicle has already formed.

---

## [Author Response]

Essential revisions:1) It would be interesting to discuss whether tumorigenic microenvironment or cellular status mimics wounding situations.

Based on the reviewers suggestion, we have now discussed this important point that is now included within two new paragraphs in the Discussion section of the revised manuscript text. Overall, we have extensively revised the Discussion text, also accommodating new results and putting important other studies in context.

2) Some parts in the manuscript including the Abstract may be misleading to provide an impression that the combination of epithelial and dermal Hh activation leads to HF, while epithelial Hh alone leads to tumor. The data shows that the Hh activation in both epidermal and dermal cells also results in BCC, but with hair follicle induction. Hair follicles are immature and abnormal and appear to be a part of the tumor. The authors should revise the text/Abstract to clearly indicate this important point.

We agree with the reviewers and we have now updated the Abstract and emphasize in the Results and in the Discussion that Hh activation in both epithelial and dermal cells also results in BCC.

3) Since Lgr6+ epithelial cells are known as stem cells, do they form hair follicles upon Hh activation in unwounded skin regardless of TD environment? In Figure 1E, Lgr6+ epithelial cells exist beyond TD. What if Hh is activated in the Lgr6+cells outside of TD and underlying stroma using Lgr6^CreER^/Col1a2^CreER^/Ptch^fl/fl^?

In the light of the reviewers comments 3, 4 and 5, we made a large effort to address the overall question whether de novo HFs form specifically in touch domes, or if de novo HFs also can form from non-touch dome areas of the interfollicular epidermis (IFE), the hair follicle isthmus/bulge, and from the hairless epidermis of the hindpaw. The experiments and respective outcomes are described in detail here below and in response to comments 4 and 5.

1) To directly address point 3, we crossed Lgr6^CreERT2^/Ptch1^fl/wt^ female with Col1a2^CreER^/Ptch1^fl/fl^ male mice (with a breeding probability of 1:8; Lgr6^CreERT2^/Ptch1^fl/wt^ males crossed with Col1a2^CreER^/Ptch1^fl/fl^ females did not result in viable litters), which gave in total four quadruple transgenic mice (Lgr6^CreERT2^/ Col1a2^CreER^/Ptch1^fl/fl^). We treated these four mice and littermate controls at postnatal week 8 (P8w) topically with 4-OH tamoxifen (TAM), and collected samples at time points post TAM treatment where we certainly would find small de novo HF buds or de novo HFs in the touch domes of Gli1^creER^/R26^Tom^/Ptch1^fl/fl^ mice. I.e. we collected dorsal skin (biopsies or entire back skin) 17 days and 31-49 days post TAM. At 17 days post TAM we observed many epithelial buds, predominantly located in the upper part of the infundibulum connecting to the IFE and occasionally in the IFE (n = 4 mice; biopsies) (Author response image 1). These epithelial buds were most often surrounded with dermal cells that expressed SDC1, similar to the dermal condensate cells in de novo HF-forming Gli1^creER^/R26^Tom^/Ptch1^fl/fl^ mice (Author response image 1). Based on this observation we expected to find plenty of de novo HFs from the IFE/infundibulum area in 31-49 post TAM skin of the same mice, which however was not the case. Either not at all (n = 2 mice) or very rarely (n = 2 mice; 1 and 2 hair shafts, respectively), we observed a small hair shaft in the IFE/infundibulum area that was likely formed de novo based on its morphology, size and thickness. However, based on HF/bulge cell morphology these hair shafts seemed not properly anchored or even soon to be shed (Author response image 1). In the touch domes of Gli1^creER^/R26^Tom^/Ptch1^fl/fl^ mice, such HFs “without proper bulge anchoring” were never observed. As a control for de novo HF formation, we also analyzed the touch domes of the Lgr6^CreERT2^/Col1a2^CreER^/Ptch1^fl/fl^ mice and indeed found in every mouse a low number of touch domes that contained de novo HFs or hair shafts (based on morphology, size and thickness) (Author response image 1). I.e. in touch domes de novo HFs likely formed but with much lower efficiency than in Gli1^creER^/R26^Tom^/Ptch1^fl/fl^ mice.

In this regard, it is important to note that *Ptch1* inactivation in the skin stroma (i.e. in Col1a2^CreER^/Ptch1^fl/fl^ mice) leads shortly after TAM injection (i.p.) to an overall bad body condition of malnutrition due to an intestinal phenotype, which is less severe but still present with topical treatment (described in the Materials and methods section of the manuscript). Unfortunately, the quadruple transgenic Lgr6^CreERT2^/Col1a2^CreER^/Ptch1^fl/fl^ developed a more severe phenotype of malnutrition than Col1a2^CreER^/Ptch1^fl/fl^, which could of course negatively impact de novo HF development.

In sum, Lgr6^CreERT2^/Col1a2^CreER^/Ptch1^fl/fl^ had numerous de novo HF-like buds at the initial time point, however the occurrence of de novo HF/hair shafts at later time points was very low or not present. As Lgr6^CreERT2^/Col1a2^CreER^/Ptch1^fl/fl^ mice did not contain a Tomato lineage tracing allele, an unequivocal confirmation that these extremely rare HFs/hair shafts were indeed newly formed was not possible, thus these experiments are not included in the revised manuscript.

**Author response image 1. respfig1:** Formation of HF-like structures in touch domes of the Lgr6/Col1a2 mouse model. (**A-D**) Lgr6^creERT2^;Col1a2^creER^;Ptch1^fl/fl^ mice were treated with Tamoxifen (TAM) at 8 weeks of age and dorsal skin was analyzed post TAM as indicated in the figure panels. Time points were chosen when de novo HF buds or HFs were clearly visible in the TDs of Gli1^creERT2^;R26^Tom^;Ptch1^fl/fl^ mice. (**A**) Formation of epithelial buds outside of the touch dome areas at 17 days post TAM treatment, predominantly located in the upper part of the infundibulum connecting to the IFE (blue and green frame) and occasionally in the IFE (yellow frame) (n = 4 mice; biopsies). (**B**) Immunofluorescent co-staining of Keratin 5 (K5) and Syndecan-1 (SDC1) of 17 days post TAM dorsal skin, showing SDC1 positive dermal cells surrounding an epithelial bud from the infundibulum region (red arrowheads). (**C**) 36 days post TAM, a small hair shaft in the IFE/infundibulum area can be observed that was likely newly formed based on its morphology, size and thickness. However, it doesn’t seem properly anchored. Inset shows a zoom-in on the indicated area, with added K5 co-staining. (**D**) In 2 touch domes de novo hair follicles or hair shafts were observed based on morphology, size and thickness (white arrowheads) at 31 or 42 days post TAM treatment, respectively (n=3 for 31-35 days post TAM, n=2 for 42-49 days post TAM). BF: bright field. Scale bars: 200um (**A**), 20um (B, C inset), 100um (C panorama), 50um (**D**).

2) The mouse hindpaw (plantar) epidermis is a skin region devoid of hair follicles and sweat glands, ideal to test whether epithelial and stromal Hh-pathway activation can induce de novo HFs from epidermal cells (i.e. outside the touch dome). Thus we probed for *Gli1* expression in the plantar skin using Gli1-LacZ reporter mice. We consistently found small *Gli1*-ΒGAL expressing clusters of epithelial and adjacent stromal cells in the plantar skin (Figure 9B, Figure 9—figure supplement 1A), which we confirmed with short-term lineage tracing in Gli1^creER^/R26^Tom^ mice (TAM at P8w; sample collection 7 days later) (Figure 9C, Figure 9—figure supplement 1B). Testing for de novo HF induction, we analyzed the hindpaws of Gli1^creER^/R26^Tom^/Ptch1^fl/fl^ mice and control littermates (Gli1^creER^/R26^Tom^/Ptch1^fl/wt^) using the same treatment scheme and sample collection times as for dorsal skin (TAM at P8w; sample collection 5 and 9 weeks post TAM). Indeed, we found numerous de novo HFs with hair shafts in the normally hairless region. These HFs were fully Tomato-lineage traced and showed normal inner-layer differentiation based on morphology and K6 staining (Figures 9D-E, Figure 9—figure supplement 2).

Taken together, these experiments in paw skin demonstrate that combined epithelial and stromal Hh-pathway activation can induce de novo HFs from epidermis of hairless skin, i.e. outside of touch dome niches. These experiments are included in the revised manuscript as new Figure 9, Figure 9—figure supplement 1 and 2 and in the subsection “Epithelial and adjacent stromal Hh-pathway activation induces de novo HF formation in hairless paw skin”.

4) Do dermal condensates form only in the stroma of TD? If this is the case, the authors may be able to show that the TD area is specialized to form hair follicles.

In Gli1^creER^/R26^Tom^/Ptch1^fl/fl^ mice the stromal cells of both touch dome and HF isthmus can be targeted (see detailed answer below in point 5). Although we observed de novo HFs in every touch dome (i.e. 100%), we also observed at very low frequency in the HF isthmus area early HF buds and de novo HFs. As shown in point 5 below, we have now added representative images and quantification of de novo HFs in the HF isthmus area, which support that de novo HFs can also form outside touch dome areas in dorsal skin.

It is interesting to note that in mice with Hh-pathway activation in the stroma (Col1a2^CreER^/ R26^Tom^/Ptch1^fl/fl^), dermal cell condensations can emerge next to the HF infundibulum as well as under the normal IFE (Figure 6G, Figure 6—figure supplement 2B), which is in support of dermal condensate formation not being restricted to the touch dome areas.

5) In Figure 1D, Gli1 is positive in follicular epithelium and surrounding stromal cells of guard hairs. When Hh is activated in Gli1/Tom/Ptch1^fl/fl^, did de novo hair follicles form in the Gli1+ area of guard hairs? It may help support that TD is a primed structure to form hair follicles in adult unwounded mouse skin.

We have now carefully assessed potential de novo HF formation in the HF isthmus areas of Gli1^creERT2^/R26^Tom^/Ptch1^fl/fl^ and control mice.

In Gli1^creERT2^/R26^Tom^/Ptch1^fl/fl^ mice we observed HFs that were most likely – based on their morphology and positioning – formed de novo HFs (Figure 8C). Unfortunately, it was not possible (as it was for de novo HFs in the touch dome) to unequivocally determine de novo HFs through lineage tracing of the infundibulum because these isthmus-associated de novo HFs seem to directly merge into the isthmus area of pre-existing HFs. However, we observed HFs in the isthmus areas with hair shafts that were thinner and often had a bent shape, which is very similar to touch dome derived de novo HFs (e.g. Figure 1I). In addition, dorsal mouse skin that is harvested at postnatal week 17 usually contains HFs with 3 hair shafts (3 rounds of anagen); however, HFs containing one of these thinner hair shafts often had in total four hair shafts (dependent on the tissue cut of 60 micrometer thickness, not all club hairs/hair shafts are present in the same section) which is an additional indication for the presence of a newly formed HF. Importantly, neither HFs with four hair shafts nor HFs with morphological de novo appearance have been observed in phenotypically normal control skin (Gli1^creERT2^/R26^Tom^/Ptch1^fl/wt^) or in skin with HF tumors (Lgr6^creERT2^/R26^Tom^/Ptch1^fl/fl^).

We quantified the frequency of de novo HFs in large skin panoramas from Gli1^creERT2^/R26^Tom^/Ptch1^fl/fl^, Gli1^creERT2^/R26^Tom^/Ptch1^fl/wt^, Lgr6^creERT2^/R26^Tom^/Ptch1^fl/wt^ and Lgr6^creERT2^/R26^Tom^/Ptch1^fl/fl^ mice (Figure 8D) and found that approx. 2% of HFs formed isthmus-associated de novo HFs. de novo HF formation was not associated with one specific hair type (e.g. guard), which is in line with the occurrence of adjacent epithelial and stromal GLI1+ cells also in the isthmus of zig-zag HFs (Figure 8A-B). Reassuringly, and as expected, we did not observe de novo HFs in skin of Lgr6^creERT2^/R26^Tom^/Ptch1^fl/fl^ mice, which lack stromal cells with high Hh signaling in the HF isthmus (*Lgr6* is not expressed in stromal cells of the HF isthmus; Füllgrabe et al., 2015).

Taken together, these results support that cells in the isthmus area (i.e. outside of touch dome niches) are competent to induce de novo HFs when high levels of Hh signaling (via Ptch1^fl/fl^) co-occur in the epithelium and adjacent stroma. These results are now included in the revised manuscript in Figure 8 and in the subsection “Cells outside TD niches can give rise to de novo HFs”.

6) Although this work clearly shows that localized Hedgehog signaling can induce hair follicles, the equivalency of these structures to existing follicles and the lack of mechanism should be addressed as detailed below.

Please find our responses to comments 7, 8 and 9 below.

7) The de novo hair follicles form under constant Hedgehog signaling, which is typically only active during the anagen phase of the hair follicle and is shut down to allow the hair follicle to proceed to catagen/telogen. Do the de novo hair follicles cycle when induced with the Gli1-CreERT2 promoter?

The de novo HFs clearly enter catagen/telogen after their first anagen (morphogenesis) as the majority of the de novo HFs in skin 9 weeks post tamoxifen are in telogen. We administered tamoxifen at 8 weeks of age. Ten days post tamoxifen we observe first de novo HF buds, 5 weeks post tamoxifen de novo HFs are in anagen (Figure 1I, Figure 2A), and 9 weeks post tamoxifen almost all de novo HFs are in telogen (Figure 2C and Figure 2—figure supplement 2). These de novo HFs have one hair shaft as this is their first telogen after morphogenesis. In the revised manuscript, we have now added additional text to better clarify this point (subsection “Characterization of de novo hair follicles in touch domes”) and added Figure 2—figure supplement 2better depicting de novo HFs in telogen 9 weeks post tamoxifen.

That de novo HFs can enter catagen/telogen despite constitutive Hh-pathway activation is not unexpected. In normal skin (i.e. during physiological hair cycling), Brownell et al. have shown that the regressing catagen follicle continued to broadly express *Gli1* using Gli1^LacZ^ reporter mice (Brownell et al., 2011; Figure 1A). With *Gli1* expression being among the most reliable direct target genes for active Hh signaling, Brownell et al. show that Hh signaling is active during catagen as well as in the telogen HF. Moreover, and most relevant for our study, HFs with continuous Hh-pathway activation via *Ptch1* inactivation can precede into catagen and telogen, which was shown in Lgr5^creERT2^/Ptch1^fl/fl^ mice where HFs entered catagen and proceeded into telogen (Kasper et al., 2011; Figure 4B, C and Supplementary Figure 6). In the follow up experiments, where Lgr5^creERT2^/Ptch1^fl/fl^ mice were tamoxifen treated at P20 and taken 13 weeks later, dorsal skin contained HFs with 3 hair shafts (instead of 1 hair shaft at P20), signifying that HFs underwent several hair growth cycles (Author response image 2).

**Author response image 2. respfig2:** H&E stained dorsal skin of Lgr5^creERT2^/Ptch1^fl/fl^ mice, treated with tamoxifen at P20. Samples were taken at P16w, when HFs normally have produced 3 hair shafts (i.e. morphogenesis and 2 times anagen). Inset shows a magnified area of left panel with black arrows indicating HFs with 3 hair shafts.

In sum, despite constitutive Hh-pathway activation via inactivation of *Ptch1,* de novo HFs do proceed from anagen into the telogen phase, which is in line with previously published work on pre-existing HFs.

8) Is Hedgehog pathway activation necessary for de novo hair follicle formation or merely Ptch1 loss?

In unwounded skin, it has been shown that activation of epithelial β-catenin in mouse skin can induce ectopic (de novo) HFs (Gat et al., 1998; Lo Celso et al., 2004). Importantly, this Wnt/β-catenin-pathway induced ectopic HF development is dependent on an active Hh-signaling pathway. Lichtenberger et al. induced ectopic HF formation using K14DNb-cateninER mice. Deletion of a single *Smo* allele in K14DNb-cateninER mice via *Dermo1Cre* (K14DNb-cateninER/Dermo^Cre^/Smo^fl/fl^) significantly reduced ectopic HF formation, and homozygous *Smo* deletion via Pdgfra^CreER^ (K14DNb-cateninER/Pdgfra^CreER^/Smo^fl/fl^) inhibited ectopic HF formation (Lichtenberger et al., 2016).

In wild type mice,de novo HF formation occurs in the center of large wounds (Ito et al., 2007). This wound-induced HF neogenesis (WIHN) is accompanied by activation of Hh/Gli signaling in the epidermal placodes and dermal condensates (Lim et al., 2018). Lim et al. furthermore demonstrated that both epithelial SHH and dermal SMO expression are necessary for WIHN as K14^creER^/Shh^fl/fl^ and Pdgfra^creER^/Smo^fl/fl^ mice, respectively, do not develop WIHN.

In the canonical Hh-pathway signaling cascade, SHH is upstream of PTCH (PTCH is the receptor for SHH) and SMO is downstream of PTCH (SMO is activated upon PTCH inhibition). Thus, the experiments of Lim et al. and Lichtenberger et al. convincingly demonstrate that for de novo HF formation active Hh/Gli-signaling is necessary. I.e. activation of the Hh-pathway via SHH/PTCH, without it’s downstream signaling component SMO that mediates Hh-pathway activation via GLI1/2 transcription factors, would not be sufficient for de novo HF formation.

In our manuscript, we have now added RNA-FISH staining for *Gli1*, which is a reliable readout for active Hh-signaling. These stainings demonstrate that de novo HF buds and established HFs within the TD express high levels of *Gli1* mRNA (i.e. have active Hh/Gli-signaling). Moreover, *Gli1* mRNA-FISH combined with antibody staining for Tomato-lineage tracing also shows that Tomato-tracing (cells with *Ptch1* deletion) and *Gli1* expression are highly correlated (Figure 4F-G, Figure 4—figure supplement 1C-D).

Taken together, in the revised manuscript we added *Gli1* mRNA-FISH staining of de novo HFs, as well as wild-type touch domes and wild-type-embryonic HF placodes/hair germs (Figure 4D-G, Figure 4—figure supplement 1). The results are described on in the subsection “De novo HF formation recapitulates embryonic HF development”.

9) Can the hair follicles persist without Hedgehog signaling?Use of Smo inhibitors such as vismodegib can be helpful to answer these questions by administering the drug after tamoxifen injection or after the hair follicle has already formed.

In the light of the reviewers comment, we administered vismodegib after the de novo HFs in touch domes had already formed. I.e. the aim of this experiment was to reveal if established de novo HFs would persist or would diminish as the BCC-tumor area does. It has been shown previously that BCC-like lesions in dorsal skin of Gli1^creER^/Ptch1^fl/fl^ shrink dramatically upon vismodegib treatment within 7 days (Eberl et al., 2018).

We treated Gli1^creER^/R26^Tom^/Ptch1^fl/fl^ mice with tamoxifen at 8 weeks. Five to seven weeks after tamoxifen treatment when de novo HFs are clearly established in the touch domes, we took a dorsal biopsy prior vismodegib treatment (i.e. untreated biopsy), and then treated the mice with vismodegib for seven days for second sample collection (i.e. 7 days Vismodegib sample). Using the optimized treatment scheme from Eberl et al., 2018 we administered 50mg/kg vismodegib i.p. once a day, for seven days (n = 4 mice) (see experimental scheme in Figure 5A).

Reassuringly, in pre-existing HFs we found considerable reduction of tumor growth as well as complete absence of Ki67 staining when comparing the untreated biopsies with 7-day vismodegib treated samples of the same mice. This reduction in tumor size demonstrated that vismodegib treatment worked well (Figure 5B-C). In touch domes, the tumor areas also dramatically diminished, however the de novo HFs persisted (Figure 5D, Figure 5—figure supplement 1). These experiments confirmed that de novo HFs indeed represent hair follicles that are independent of tumor structures as they persist upon vismodegib treatment when the surrounding BCC-growth areas are nearly gone.

Taken together, we added these new experiments to the revised manuscript (Figure 5, Figure 5—figure supplement 1) and updated the manuscript text accordingly (subsection “De novo HFs persist while BCC-like tumors diminish upon vismodegib treatment”).